JCB Journal of Cell Biology

## REPORT

# Nanoscale structural organization and stoichiometry of the budding yeast kinetochore

Konstanty Cieslinski[1,2]*, Yu-Le Wu[1,3]*, Lisa Nechyporenko[1,4], Sarah Janice Hörner[1,5,6], Duccio Conti[7], Michal Skruzny[1], and Jonas Ries[1]

**Proper chromosome segregation is crucial for cell division. In eukaryotes, this is achieved by the kinetochore, an evolutionarily conserved multiprotein complex that physically links the DNA to spindle microtubules and takes an active role in monitoring and correcting erroneous spindle–chromosome attachments. Our mechanistic understanding of these functions and how they ensure an error-free outcome of mitosis is still limited, partly because we lack a complete understanding of the kinetochore structure in the cell. In this study, we use single-molecule localization microscopy to visualize individual kinetochore complexes in situ in budding yeast. For major kinetochore proteins, we measured their abundance and position within the metaphase kinetochore. Based on this comprehensive dataset, we propose a quantitative model of the budding yeast kinetochore. While confirming many aspects of previous reports based on bulk imaging, our results present a unifying nanoscale model of the kinetochore in budding yeast.**

## Introduction

Cell division is a process of paramount importance for organismal life, ultimately ensuring the faithful propagation of the genome in space and time. During mitosis, the kinetochore, an architecture-conserved multiprotein complex across all eukaryotes (Drinnenberg et al., 2016; van Hooff et al., 2017), takes part in several key processes (Joglekar et al., 2010; Aravamudhan et al., 2015; Asbury, 2017). It contributes to appropriate chromosome segregation, which prevents an aberrant karyotype and thus subsequent developmental defects or cell death (Santaguida and Amon, 2015). The kinetochore assembles at the centromere of each sister chromatid to generate robust connections between the chromatin and spindle microtubules. It serves as a platform for the spindle assembly checkpoint proteins and senses pulling forces between the chromosomes and the mitotic spindle (reviewed in Musacchio and Desai, 2017). A simple model for studying the complex is the budding yeast, *Saccharomyces cerevisiae*, where the kinetochore assembles onto one nucleosome and is attached to one microtubule (Winey et al., 1995). Conversely, multiple copies of units analogous to the budding yeast kinetochore bind to many microtubules in other fungi and multicellular organisms (Zinkowski et al., 1991; Musacchio and Desai, 2017). The functions of the kinetochore are strongly dependent on its structure and potential remodeling over the cell cycle (Joglekar et al., 2009; Conti et al., 2017; Dhatchinamoorthy et al., 2017).

Early EM studies defined three electron-dense regions in the kinetochore—the inner kinetochore, the outer kinetochore, and the fibrous corona (Rieder, 1982). In *S. cerevisiae*, where the corona is absent, the inner kinetochore includes the centromeric nucleosome containing an H3 variant called Cse4, the CBF3 complex (Cep3, Ndc10, Ctf13, and Skp1), the Mif2 and Cnn1 module (Cnn1, Wip1, and Mhf1/2), the Mcm16/Ctf3/Mmc22 complex, Nkp1/2, the COMA complex (Ctf19, Okp1, Mcm21, Ame1), and the Chl4/Iml3 dimer. The outer kinetochore consists of the microtubule-interacting network built by Spc105, the MIND complex (Mtw1, Dsn1, Nnf1, and Nsl1), the Ndc80 complex (Ndc80c; Ndc80, Spc24, Spc25, and Nuf2), and the Dam1 complex (Dam1c; including Dam1 and Ask1) ring (Musacchio and Desai, 2017; Fig. 1 A).

Despite advances in the last decades in understanding kinetochore composition, a complete picture of its organization in cells is still unclear. Many structures of the kinetochore components in both human and budding yeast have been solved (for an overview, see Dimitrova et al., 2016; Musacchio and Desai, 2017; Jenni and Harrison, 2018; Hinshaw and Harrison, 2019; Yan et al., 2019; Pesenti et al., 2022; Yatskevich et al., 2022). EM

[1]Cell Biology and Biophysics Unit, European Molecular Biology Laboratory, Heidelberg, Germany; [2]Translational Radiation Oncology Unit, Deutsches Krebsforschungszentrum, Heidelberg, Germany; [3]Faculty of Biosciences, Collaboration for Joint PhD Degree Between European Molecular Biology Laboratory and Heidelberg University, Heidelberg, Germany; [4]Institute of Pharmacy and Molecular Biotechnology, Heidelberg University, Heidelberg, Germany; [5]Institute of Molecular and Cell Biology, Mannheim University of Applied Sciences, Mannheim, Germany; [6]Interdisciplinary Center for Neuroscience, Heidelberg University, Heidelberg, Germany; [7]Department of Mechanistic Cell Biology, Max Planck Institute of Molecular Physiology, Dortmund, Germany.

*K. Cieslinski and Y.-L. Wu contributed equally to this paper.   Correspondence to Jonas Ries: jonas.ries@embl.de.

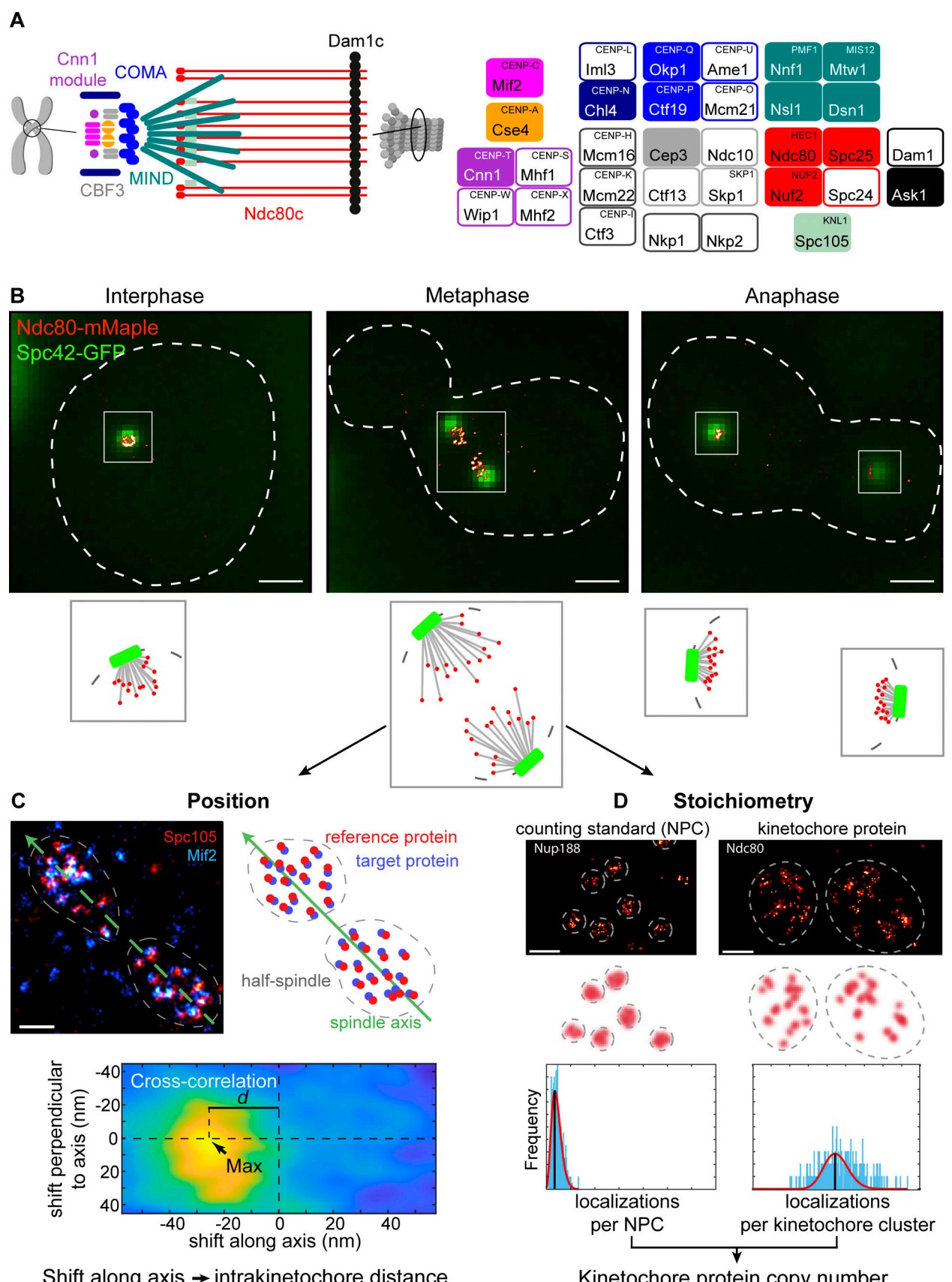

Figure 1. **Overview of the study. (A)** Protein composition of the budding yeast kinetochore. Kinetochore proteins are grouped and color-coded by complexes. Only opaquely colored components were measured in this study. Human counterparts are shown in a superscript. Note that this is not an exhaustive

list. **(B)** Example kinetochore clusters. Overlays of representative super-resolved images of the kinetochore protein Ndc80 (red) and the diffraction-limited images of spindle pole body protein Spc42 (green) at different stages of the cell cycle and corresponding cartoons of the budding yeast spindles. Scale bars: 1 µm. **(C)** The position of kinetochore proteins along the spindle axis. We always labeled and imaged the reference protein Spc105 (red) together with the target protein (cyan, Mif2 in this example). We manually segmented single kinetochore clusters, defined the spindle axis, and calculated the image cross-correlation. The position of the cross-correlation peak corresponds to the average distance between reference and target proteins in the half spindle. Scale bar: 200 nm. **(D)** Stoichiometry of the budding yeast kinetochore proteins. We quantified the copy numbers of kinetochore proteins using the NPC component Nup188, which has 16 copies per NPC, as a counting reference standard. In each experiment, we mixed two strains in which either Nup188 or the target kinetochore protein were labeled with the same fluorescence protein tag mMaple. We then imaged both strains simultaneously. We calculated the ratio of mean localization counts per structural unit (either NPC or kinetochore cluster) between the two proteins. From the relative number of localizations and the known stoichiometry of Nup188, we computed the copy number of the target kinetochore protein. Scale bars: 200 nm.

studies have revealed the overall shape of the kinetochore (Gonen et al., 2012; McIntosh et al., 2013), but lack the molecular specificity to position most proteins (Hinshaw and Harrison, 2019; Yan et al., 2019). Meanwhile, conventional fluorescence microscopy has molecular specificity (Joglekar et al., 2009; Haase et al., 2013; Aravamudhan et al., 2014) but has an insufficient resolution (Abbe, 1873) to separate individual complexes. As a result, in budding yeast, all 16 kinetochores appear as one and two fluorescent spots during interphase and mitosis, respectively (Joglekar et al., 2006), lacking fine structural details. Thus, a comprehensive in situ structural understanding of the kinetochore is still missing.

In the budding yeast kinetochore, built on a short centromere sequence (~125 bps; Clarke and Carbon, 1980), the microtubule is captured by a Dam1c ring and several copies of Ndc80c. Precisely how many complexes are present remains controversial. Fluorescence microscopy has been used to quantify the absolute copy numbers of the major kinetochore components based on a reference protein with a known copy number (Joglekar et al., 2006, 2008; Lawrimore et al., 2011; Dhatchinamoorthy et al., 2017). These studies generally agreed that the outer kinetochore proteins are more abundant than the inner kinetochore proteins. Ndc80 has been shown to be present in 6–19 copies per kinetochore. Smaller or equal amounts were found for the MIND complex (4–7 copies) and Spc105 (4–5 copies). The COMA complex was shown to be present in 2–5 copies. Within the inner kinetochore, Cep3 was found to have 2–3.4 copies, Mif2 2–3.6 copies, and Cnn1 and Cse4 2–6 copies (Shivaraju et al., 2012; Wisniewski et al., 2014). Such large discrepancies may arise from differing experimental conditions (Joglekar et al., 2008) and prevent generating a detailed structural model. Open fundamental questions include: How do the Mif2 and Cnn1 assembly pathways quantitatively contribute to the copy number of Ndc80c? How many COMA complexes exist within the budding yeast kinetochore?

Another extensively debated question in the field is the exact stoichiometry of the histone protein Cse4 at centromeres (Clarke and Carbon, 1980; Ng and Carbon, 1987; Keith and Fitzgerald-Hayes, 2000). To date, a series of alternative structures have been proposed to define the nature of the centromeric nucleosome. These hypotheses include hemisome (Bui et al., 2012; Dalal et al., 2007), hexameric (Mizuguchi et al., 2007), or octameric configurations (Camahort et al., 2009), where a single or two copies of Cse4 are present (Black and Cleveland, 2011). With regards to the Cse4 copy number, biochemical approaches have reported the presence of a single Cse4 nucleosome at centromeres (Furuyama and Biggins, 2007; Krassovsky et al., 2012).

In contrast, in vivo studies showed a high variability of Cse4 copy number per kinetochore, ranging from 2 (Shivaraju et al., 2012; Wisniewski et al., 2014; Dhatchinamoorthy et al., 2017) up to 4–6 copies (Lawrimore et al., 2011). Therefore, the identity and copy number of the centromeric nucleosome remain unanswered.

Super-resolution microscopy, specifically single-molecule localization microscopy (SMLM; Betzig et al., 2006; Hess et al., 2006; Rust et al., 2006), achieves nanometer resolution combined with molecular specificity. It has been used to gain structural insights into the organization of multiprotein complexes such as the nuclear pore complex (Szymborska et al., 2013), the endocytic machinery (Sochacki et al., 2017; Mund et al., 2018), centrioles (Sieben et al., 2018), or synaptic proteins (Dani et al., 2010). Here, we use SMLM to determine the location of key proteins and their copy numbers with single kinetochore resolution in *S. cerevisiae* cells (Fig. 1). From these data, we built a comprehensive model of how the major components are positioned and what their stoichiometry is in the budding yeast metaphase kinetochore in situ.

## Results and discussion

### Individual kinetochores can be observed with SMLM

To determine whether SMLM can be used to visualize individual kinetochores, we imaged yeast cells in which Ndc80 was endogenously tagged with mMaple, and Spc42 (spindle pole body protein) with GFP (Fig. 1 B). When we imaged unsynchronized cells, we observed that in interphase cells, all kinetochores are packed within a small cluster with a size below the resolution limit of standard microscopy, with the tendency to organize into a rosette-like configuration similar to what is observed in human cells in early prometaphase (Fig. 1 B; Chaly and Brown, 1988; Jin et al., 2000; Bystricky et al., 2005). In metaphase, kinetochores did not generate a metaphase plate but rather organized into two sister kinetochore clusters (Fig. 1 B). In late mitosis, the separation of the sister kinetochore clusters increases (Fig. 1 B; Joglekar et al., 2006). At this late stage of division, their high density did not allow us to resolve individual kinetochores with SMLM. In conclusion, SMLM allows visualizing single kinetochores within the budding yeast spindle in interphase and metaphase.

### Kinetochore subunits are organized functionally along the metaphase spindle axis

To resolve the structural details of the kinetochore, we used dual-color super-resolution imaging to map kinetochore proteins

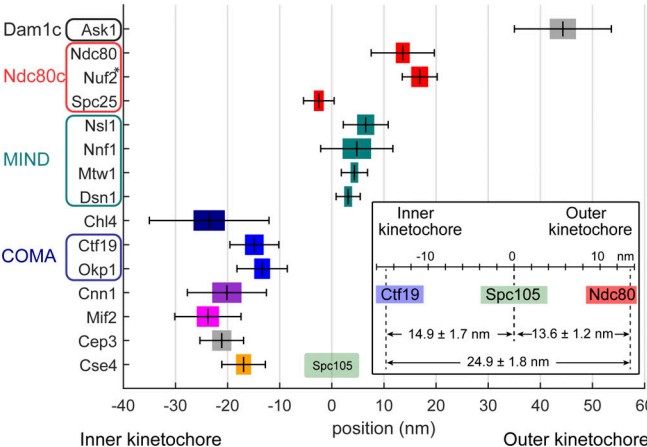

**Figure 2. Position of 15 kinetochore proteins along the spindle axis with Spc105 as a reference point.** All proteins were tagged at their C-termini. The mean distance is plotted with the SEM (colored box) and SD (whiskers). The inset depicts control measurements showing consistency in pairwise distance measurements ± SEM among three proteins. See Table 1 for values and sample size. *The position of Nuf2 is based on the measured pair Ndc80–Nuf2.

onto the spindle axis in a single dimension. Here, we focused on essential kinetochore components, improved the positioning accuracy of some proteins (Joglekar et al., 2009), and added new position information on the Cse4 C-terminus, Cep3, Mif2, Cnn1, and Chl4. We imaged the spatial reference (Spc105 unless indicated otherwise) and the target proteins in separate channels. We analyzed each kinetochore cluster individually and determined the relative spatial offset between the two channels by cross-correlation as the target-reference distance (Fig. 1 C and Fig. S1, and Materials and methods). The possible tilt of the spindle axis introduces only a minimal measurement error (maximum error = 6.3%, mean error = 2.1%; see Fig. S2 and Materials and methods). We only analyzed metaphase cells where both kinetochore clusters allowed for high-quality position measurements. With this, we precisely determined the pairwise distances between 15 pairs of kinetochore proteins, all labeled at their C-termini (Fig. 2). We further validated this approach with an independent analysis measuring individual kinetochores (Fig. S3 A) and obtained highly similar results. Our measurements of different kinetochore proteins were internally consistent, as the sum of the measured Ndc80–Spc105 (13.6 ± 1.2 nm; mean ± SEM) and Spc105–Ctf19 (14.9 ± 1.7 nm) distances is close to the measured Ndc80–Ctf19 distance (24.9 ± 1.8 nm; Fig. 2 inset). These data agree reasonably well with previous diffraction-limited dual-color microscopy studies with the noticeable exception of MIND components position (for comparison, see Table S1 and Fig. S3 B; Joglekar et al., 2009). Furthermore, we found that the C-termini of Ndc80 and Nuf2 are in close proximity, with a distance of 3.3 ± 1.5 nm (Fig. 2), which agrees well with a distance of 3.6 nm, as determined from a crystal structure (Valverde et al., 2016), adding another validation. In summary, these data show that SMLM dual-color imaging is suitable to measure intrakinetochore protein distances in budding yeast.

Our results show that, within the centromere-proximal region, Cse4 and CBF3 (measured with its constituent Cep3) co-localize with each other as well as with the C-termini of Chl4 and of both Mif2 and Cnn1, which are outer kinetochore receptors. Specifically, we found that the C-termini of Cse4 and Cep3 are positioned within 1.5 nm at the centromeric site. Also, Mif2 and Cnn1 cluster together, which is consistent with their function within the inner kinetochore (Fig. 2), but are around 3 nm away from the Cse4, toward the Cep3 site. We find that Ctf19 and Okp1 (COMA components) are –14.9 ± 1.7 and –13.4 ± 1.4 nm away from Spc105, respectively, toward the centromere (Fig. 2). Indeed, the Cep3 dimer has been shown to bind CDEIII DNA and participates in Cse4-containing centromere deposition (Leber et al., 2018; Yan et al., 2018; Zhang et al., 2018; Hinshaw and Harrison, 2019). Cnn1 does not seem to directly bind the centromeric nucleosome but its localization depends on Mif2 (Schmitzberger et al., 2017). Interestingly, we measured the position of Chl4 to be only 0.3 nm away from Mif2, but more distant from the COMA complex (8.9 nm), which occupies the intermediate position (15–20 nm from Spc105), thus bridging the inner with the outer kinetochore (Hornung et al., 2014; Hinshaw and Harrison, 2019). This finding is in line with Chl4 interacting with Mif2, the Cse4-containing nucleosome and, electrostatically, with DNA (McKinley et al., 2015; Pentakota et al., 2017).

Next, we find that the outer kinetochore components (Spc105, MIND, Ndc80c, and Dam1c) are more distal from the centromere. For example, Nnf1, Nsl1, Mtw1, and Dsn1 of the MIND complex are located between 3.1 and 6.5 nm away from Spc105 in the outward direction (toward the microtubule). This is consistent with a crystal structure of MIND in yeast and humans and with the known binding site of the KNL1$^{\text{Spc105}}$ C-terminus on the MIND complex (Hornung et al., 2014; Kudalkar et al., 2015; Petrovic et al., 2014; Dimitrova et al., 2016; Petrovic et al., 2016). Here, the C-terminus of Dsn1 highly overlaps with the Spc105 position, whereas Nnf1, Mtw1, and Nsl1 C-termini extend toward the position of Ndc80. This adjusts a previous diffraction-limited microscopy study that measured the distance in living cells and found the C-termini of Mtw1, Nsl1, and Dsn1 7 nm away from Spc105 toward the centromere, whereas Nnf1 fully colocalized with Spc105 (Joglekar et al., 2009). While the C-terminus of Spc25 is adjacent to the C-termini of both Spc105 and MIND (Fig. 2), the C-terminus of Ndc80 occupies a more outward position. All C-termini of the MIND complex are localized more than 10 nm away from COMA, suggesting that all N-terminal regions of MIND proteins lie relatively close to the complex. This is supported by numerous previous biochemical and optical studies (Aravamudhan et al., 2014; Dimitrova et al., 2016; Petrovic et al., 2016). The distance between the position of COMA and the C-termini of MIND implies a possible tilt between the long axis of MIND and the spindle as the total length of Mtw1 is around 20 nm (Hornung et al., 2011). The structured segment of Spc105, the reference point, is positioned close to the C-termini of MIND, as was proposed previously using structural approaches (Petrovic et al., 2014). The Ndc80c is an elongated heterotetramer. The C-termini of two of its constituents (Spc25

and Ndc80) are 14.1 nm away from each other, a few nanometers less than the maximum length of this region observed in purified proteins (Wei et al., 2005; Valverde et al., 2016). The discrepancy between the structural data of MIND and Ndc80c in our measurements can be explained as an existing tilt of both complexes to the spindle axis under attachment tension. Based on the distance between Okp1 and Ndc80, the tilt can be estimated to be around 46°. Finally, the Ask1 subunit of Dam1c, which also binds the microtubule surface (Cheeseman et al., 2001; Miranda et al., 2005; Westermann et al., 2005), is positioned around 40 nm away from Spc105 in the microtubule direction.

Our data also contain information about the distribution widths of kinetochore proteins perpendicular to the spindle axis. We extracted this information using autocorrelation analysis. We found that the width of the distribution correlates to the position of the protein along the spindle axis (Fig. S4). Using autocorrelation of simulated ring distributions with different radii as references, we found that most inner kinetochore proteins are distributed within a radius of 10–15 nm of the kinetochore center and most outer kinetochore proteins within a radius of ~15 nm. The wider distributions of the outer kinetochore proteins can be explained by the presence of a microtubule, which has a radius of ~12.5 nm, occupying the central space.

In summary, we mapped the relative positions of 16 kinetochore proteins along the spindle axis with nanometer precision. Generally, our results align with previous biochemical complex reconstitutions, protein interaction studies, and most optics-based distance measurements. The resulting position map clearly showed that the structural organization of kinetochore proteins correlated with their function and confirmed the general structure of the inner and the outer kinetochore. In our analysis, we found kinetochore proteins known to interact with each other in close proximity, validating their interactions and our approach.

### Counting copy numbers of major kinetochore components with quantitative SMLM

To estimate the protein copy numbers of the major kinetochore components (targets), we performed molecular counting using a reference standard (Thevathasan et al., 2019). The copy number of a target was simply calculated from the relative number of detected localizations, given that the copy number of the reference is known. Here, the target and the reference were genetically tagged with the same fluorophore (mMaple) in different strains and imaged under identical conditions on the same coverslip. The reference strain can be identified by the marker Abp1-GFP. For the reference, we selected the nucleoporin Nup188, which has 16 copies in one budding yeast nuclear pore complex (NPC; Kim et al., 2018), which is bright and easy to segment (Fig. 1 C; Thevathasan et al., 2019). We only analyzed kinetochore clusters that were close to the focal plane to ensure that the analyzed kinetochore proteins did not exceed the imaging depth (see Fig. S2 and Fig. S5 A and Materials and methods) and to precisely calculate the copy numbers.

Different lifetimes of the Nup188 and kinetochore proteins could lead to different maturation efficiencies of the mMaple tag

and consequently to systematic errors in the counting measurements. To investigate the effect of tag maturation, we transiently stopped protein translation with 250 μg/ml cycloheximide (CHX) and performed our counting measurements 1 h after this treatment (Fig. S5 B). Although we observed minor changes in copy numbers, the overall effect of CHX was small except for internally tagged Cse4. We have not noticed any growth defects that may have arisen from the tagging in our experiments, but we do not exclude the possibility of minor effects. However, our data is consistent with the previous measurements suggesting that our C-terminal tagging did not introduce any artifacts (Joglekar et al., 2006; Joglekar et al., 2008; Joglekar et al., 2009; Lawrimore et al., 2011; Pekgöz Altunkaya et al., 2016; Dhatchinamoorthy et al., 2017). We conclude that tag maturation does not affect our measurements of protein copy number.

One highly debated question in the centromere field is the composition of the centromeric nucleosome and the copy number of Cse4 within individual kinetochores. Using biochemical assays such as chromatin immunoprecipitation, only a single centromere-specific nucleosome can be recovered (two Cse4 copies; Furuyama and Biggins, 2007; Krassovsky et al., 2012; Pekgöz Altunkaya et al., 2016), while using in situ assays such as fluorescence microscopy, higher copy numbers have been reported (Lawrimore et al., 2011). In our experiments, we found Cse4 in 4.8 ± 2.4 (mean ± SD) copies (Fig. 3) when the tag is placed at its C-terminus. Previous reports have indicated that N- or C-terminal tagging of Cse4 renders cells less viable (Wisniewski et al., 2014) than internal tagging, which is compatible with its physiological function. However, we found that both internal (near the N-terminus) and C-terminal tagging of Cse4 were compatible with viability, and their copy numbers were essentially identical, with 4.2 ± 2.0 copies of the histone Cse4 when it is tagged internally. Also, we observed 30–40% reduction in the copy number upon CHX treatment (Fig. S5 B). Given that non-centromeric Cse4 has been shown to turnover continuously to ensure only one centromere per chromosome (Collins et al., 2004; Krassovsky et al., 2012), this reduction could reflect a decreased concentration of Cse4.

To obtain further information about the centromere environment, we measured the copy numbers of the Cse4 binders Mif2 and Cep3 (CBF3 complex). Cep3 was found in four (4.2 ± 2.1) copies, with an equal copy number to Cse4. Four copies (3.5 ± 1.7) of Mif2 were found per kinetochore, showing that Mif2 may be present as two dimers. Cnn1 is present in 2.1 ± 1.3 copies/kinetochore. The CBF3 complex containing two Cep3 dimers was shown to potentially allocate to a kinetochore (Yan et al., 2018). However, Cep3 also exhibits non-kinetochore localization (Joglekar et al., 2006). Note that in other organisms, the Mif2-[CENP-C] dimer can interact with two centromeric nucleosomes distinguishing the budding yeast centromere even more (Carroll et al., 2010; Guse et al., 2011; Watanabe et al., 2019; Ali-Ahmad et al., 2019; Walstein et al., 2021). Our study supports the notion that, among other inner-kinetochore components, non-centromeric Cse4 may play a role in maintaining the "point" centromere by serving as a spare module (as discussed in Scott and Bloom, 2014). We found four (4.1 ± 1.9) copies of the

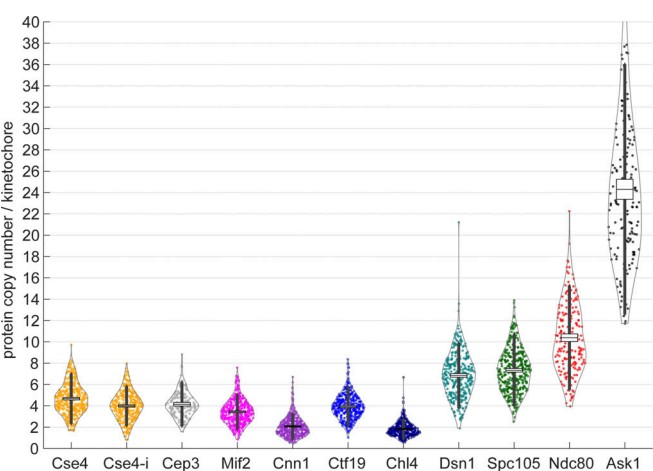

**Figure 3. Protein copy numbers per kinetochore measured with Nup188-mMaple as a counting reference standard.** Each data point corresponds to one kinetochore cluster. All proteins were tagged at their C-termini, except Cse4-i that was tagged internally. Boxes denote average copy numbers and SEMs, and whiskers denote SDs. For each protein, two independent experiments were performed and pooled. The pooled copy number and standard deviation were calculated as $\bar{N}_k = \sum_j (N_{ki}M_{ki})/\sum_i M_{ki}$ and $\bar{S} = \sqrt{\sum_i \left((M_{ki}-1)S_i^2\right)/\left(\sum_i M_{ki} - I\right)}$, respectively. Here, $N_{ki}$, $M_{ki}$, and $S_i$ are the copy number, sample size, and SD of the $i^{th}$ of total $I = 2$ replicates, respectively. The pooled SEM was given by $\bar{S}_m = \bar{S}/\sqrt{\sum_i M_{ki}}$ (see Materials and methods for details). Sample size: 389 (Cse4), 347 (Cse4-i), 157 (Cep3), 397 (Mif2), 378 (Cnn1), 357 (Ctf19), 362 (Chl4), 317 (Dsn1), 387 (Spc105), 183 (Ndc80), 156 (Ask1) kinetochore clusters, from two replicates each.

COMA complex component Ctf19 but only two (1.8 ± 1.0) copies of Chl4, a COMA and Mif2 binder, per kinetochore. Structural studies have shown only two COMA complexes within a kinetochore (Hinshaw and Harrison, 2019). As the human CCAN[COMA] complex can stably bind DNA in the absence of Cse4[CENP-A] (Pesenti et al., 2022; Yatskevich et al., 2022), we thus suggested that additional COMA copies as accessory (non-centromeric). It is widely accepted that N-termini of both Mif2 protein and COMA subunits allow and regulate the assembly of the outer kinetochore module (Przewloka et al., 2011; Screpanti et al., 2011; Dimitrova et al., 2016; Petrovic et al., 2016). With a total of two interaction sites from a Mif2 dimer and two COMA, a budding yeast kinetochore may build up to four copies of MIND.

The outer kinetochore proteins are present in higher copy numbers: 7.6 ± 3.4 copies of Spc105, 7.2 ± 3.2 of Dsn1, 10.9 ± 5.0 of Ndc80, and 24.9 ± 11.0 of Ask1 (Fig. 3). The higher copy number of Dsn1 (MIND complex) than the interaction sites would leave additional copies unbound. However, the crystallographic packing of MIND reveals potential oligomerization (Dimitrova et al., 2016), allowing us to place all complexes within the kinetochore. This in turn would bring an equal or similar amount of Spc105 and MIND complexes (Petrovic et al., 2014), in line with our finding. Consistent with others (Joglekar et al., 2006; Dhatchinamoorthy et al., 2017), we found more Ndc80 than Spc105 or MIND per kinetochore. We have estimated a slightly higher copy number of Ask1 protein (one copy per Dam1c

monomer) than an earlier work (16–20 copies; Joglekar et al., 2006). Although 17 Dam1c monomers were shown to form a complete microtubule-encircling ring (Ng et al., 2019), different configurations of the oligomerization (one and two partial/complete rings) might exist, even in the same cell (Kim et al., 2017; Ng et al., 2019). These configurations may explain the variation and higher mean copy number of Ask1 we quantified.

The estimated copy number ratio of Ndc80 to Cse4 in the current analysis is 2.5, different from the ratio of four reported by the aforementioned studies. The additional two Ndc80 can be bound by the outer kinetochore receptor Cnn1, which has been shown to bind two to three Ndc80c (Huis in 't Veld et al., 2016; Pekgöz Altunkaya et al., 2016). The decreasing activity of kinases that can regulate the binding (e.g., Cdk1 and Mps1; Malvezzi et al., 2013) may allow the Cnn1–Ndc80 interaction to be more permissive. Yet, when Ndc80c copy numbers are estimated in Cnn1-deleted strains, the copy number is not altered (Pekgöz Altunkaya et al., 2016; Dhatchinamoorthy et al., 2017) or the change may be minimal when MIND–Ndc80c binding pathway is impaired (Lang et al., 2018). This points to the redundancy of Cnn1 in budding yeast when the mitotic checkpoint is not compromised or to dynamic nature of the Ndc80–Cnn1 interaction.

**Quantitative model of the budding yeast kinetochore**
By integrating all the high-accuracy protein copy numbers (Fig. 3) and protein–protein distance measurements along the spindle axis (Fig. 2), we obtained a comprehensive model of the structural organization of the budding yeast kinetochore (Fig. 4), revising previous models (Jenni et al., 2017; Fischböck-Halwachs et al., 2019; Hamilton et al., 2019; Ustinov et al., 2020). Based on the proteins' close proximity (Fig. 2), their reported dimerization (Cohen et al., 2008), and non-centromeric DNA interactions, we positioned at the centromeric site two copies of Cse4, a dimeric CBF3 subunit (with two Cep3 dimers), Mif2 dimer, and two copies of Cnn1. Roles of the additional copies of Cse4, Mif2, CBF3, and COMA molecules detected by our measurements (indicated in Fig. 4 by dashed lines) need to be further investigated. In addition, we only included essential structural information (protein structure and binding partners) well-established in the field. Specifically, we did not divide the inner-kinetochore components by their centromeric-proximal, peri-centromeric, or other nuclear localization. Next, we placed all C-termini of MIND proteins away from COMA. We then positioned seven copies of Spc105 and MIND and 10 globular Spc25-containing ends of Ndc80c near each other. Four unbound Ndc80c were left for Cnn1 binding. Finally, we present Dam1 complexes as an oligomeric structure surrounding the microtubule. Our model adds valuable information to understand how the budding yeast metaphase kinetochore is structurally organized in situ by overcoming the resolution limit present in the previous studies.

In an independent investigation, a similar methodology was used to assess the protein composition and distances of *S. pombe* kinetochores (Virant et al., 2021 Preprint). Their results are in excellent agreement with ours, as expected from the high conservation of kinetochore components across the two yeast species (van Hooff et al., 2017), validating our respective

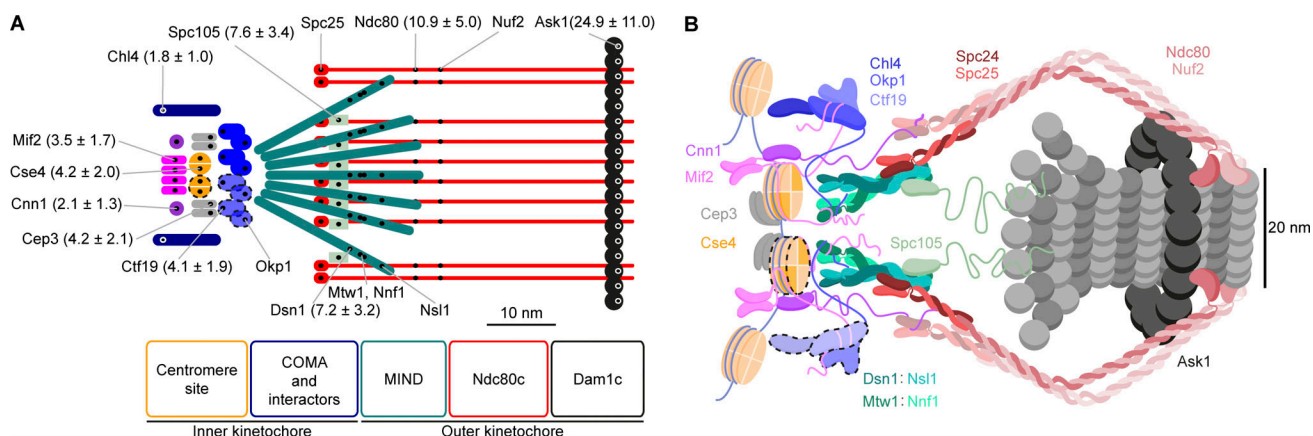

Figure 4. **Structural model of the budding yeast kinetochore. (A)** Quantitative schematic model based on the position and protein copy numbers measured with SMLM. The position of the label is shown as a small black dot. Values in the parentheses are the estimates of the number of proteins per kinetochore ± SD. **(B)** Illustrative structural model that we built by integrating our position and copy number measurements with previous models (Jenni et al., 2017; Fischböck-Halwachs et al., 2019; Hamilton et al., 2019; Ustinov et al., 2020). Dashed lines indicate potentially accessory (non-centromeric) copies (see Results and discussion for details). For simplicity, only two copies of COMA, MIND, and Spc105 and four copies of Ndc80c are shown in B.

approaches. One main difference is the Cse4:COMA ratio, which is 1:0.9 in budding yeast and 1:2.1 in fission yeast, pointing to intrinsic stoichiometry changes between point and regional kinetochores. In conclusion, our quantitative SMLM methods provide a strong basis for future studies, for instance how kinetochore components are organized perpendicular to the spindle axis and how this relates to the kinetochore–microtubule binding management, how their structure and stoichiometry change throughout the cell cycle, or how kinetochores are organized in other organisms. Our methods are not restricted to kinetochores but will enable quantitative measurements of the stoichiometry and structure of other multiprotein assemblies in situ.

## Materials and methods

### Yeast strain generation
All strains used in the study (Table S2) were derived from *S. cerevisiae* MKY0100 strain (S288c derivative), a kind gift from the Kaksonen lab (University of Geneva, Geneva, Switzerland). The strains for endogenous expression of fluorescently tagged kinetochore proteins were created by homologous recombination using PCR-based C-terminal tagging cassettes (Janke et al., 2004). The cassettes were created by amplification of DNA regions of respective pFA6a plasmids (Mund et al., 2018) encoding mMaple (McEvoy et al., 2012) or SNAP$_f$ tag (Sun et al., 2011) with S3 and S2 primer pairs (Table S3). In brief, 100 ng of the plasmid was combined with 10 μl of Hi-Fi Reaction Buffer (Bioline), 0.25 mM dNTPs (Bioline), 0.5 mM of each primer, 0.2 U of Velocity polymerase (Bioline), and 2 mM MgCl$_2$ solution (Bioline) in a total volume of 50 μl. The reaction was performed with the following steps: 98°C for 2 min, followed by 35 thermal cycles (98°C for 30 s, 56°C for 30 s, and 72°C for 90 s), and 72°C for 3 min. The Cse4–mMaple–Cse4 strain was created analogically to Wisniewski et al. (2014). Cse4 and mMaple sequences were amplified by PCR and ligated into pFA6a vector replacing a tag sequence.

PCR products were used to transform yeast competent cells by standard lithium–acetate protocol. For the transformation reaction, 15 μl of unpurified PCR product was gently mixed with 50 μl of the competent cells, 360 μl of PEG buffer (100 mM lithium acetate, 10 mM Tris-HCl, pH 8, 1 mM EDTA, pH 8, and 4% wt/vol PEG 3350 in H$_2$O) and left at room temperature for 30 min. 47 μl of DMSO were added to the reaction followed by 15 min incubation at 42°C. After heat shock, the cells were spun down at 350 × *g* for 2.5 min. The pellet was resuspended in water and plated onto a selection plate. The plate was then incubated at 30°C for 3 d.

Correct genome integrations in transformed yeast cells were checked by colony PCR with a forward primer ("frw check" ones in Table S3) annealing inside the cassette at the 3′ end and a reverse one ("rev check" ones in Table S3) that binds a sequence ~200 bp downstream of the gene of interest. The reaction was assembled by adding a small amount of a colony to 20 μl of 1× MangoMix (Bioline) mixed with 0.5 mM of each primer. The PCR reaction was performed under the following conditions: 95°C for 10 min, followed by 30 thermal cycles (95°C for 30 s, 51°C for 30 s, and 72°C for 2 min), and 72°C for 4 min.

### Sample preparation
24-mm round coverslips were cleaned in HCl/Methanol overnight and then rinsed with water. Additionally, the coverslips were cleaned using a plasma cleaner to remove residual organic contaminations. Coverslips were then coated with 15 μl of Concanavalin A (4 mg/ml in PBS; C2010; Sigma-Aldrich), dried overnight at 37°C, and rinsed before use with water to remove residual PBS. The coverslip was covered with ~100 μl of a cell suspension and incubated for 15 min.

For mMaple imaging, 2 ml of yeast logarithmic culture was grown in SC-Trp, spun down (2,500 rpm, 3 min), and resuspended in 100 μl of the medium. In the case of control experiments with CHX treatment, 250 μg/ml of CHX (in DMSO) was added to cells 1 h before immobilization. Cells immobilized on Concanavalin A–coated coverslips were fixed in 4%

paraformaldehyde and 2% sucrose in PBS for 15 min at room temperature. Fixation was quenched by two washes in 100 mM ammonium chloride, pH 7.5, in PBS for 20 min. Finally, the sample was rinsed with PBS several times. The coverslip was mounted on a microscope stage and covered with 50 mM Tris-HCl, pH 8, in 95% $D_2O$.

For single- and dual-color imaging with SNAP, the cells were immobilized, fixed, and washed the same way. Subsequently, the cells were permeabilized by 0.01% digitonin in 1% BSA solution for 30 min at room temperature under moist conditions. The sample was then washed in PBS. The sample was labeled with 1 µM SNAP-Surface Alexa Fluor 647 (S9136S, New England Biolabs) in 1% BSA solution for 2 h at room temperature under moist conditions. Finally, the sample was washed in PBS 3 × 5 min. The sample was mounted in a microscope stage and covered with the blinking buffer consisting of 50 mM Tris-HCl, pH 8, 10 mM NaCl, 10% (wt/vol) D-glucose, 500 µg/ml glucose oxidase, and 40 µg/ml catalase in 90% $D_2O$ (Thevathasan et al., 2019). The blinking buffer for Alexa Fluor single-color or dual-color imaging was supplemented with 35 mM or 15 mM MEA (mercaptoethylamine), respectively.

### Microscopy

The SMLM acquisitions were performed with two equivalent custom-built microscopes, analogically as in Mund et al. (2018). The microscopes were controlled with Micro-Manager (Edelstein et al., 2014) using EMU, a custom interface (Deschamps and Ries, 2020). Both microscopes have several available channels/colors—UV (405 nm), green (488 nm laser, 525/50 nm emission bandpass filter), orange (561 nm laser, 600/60 nm emission bandpass filter), and red (640 nm—excitation and booster laser, 700/100 nm emission bandpass filter). A focus lock system based on a totally reflected IR laser beam was used to keep the focus constant. The UV laser was adjusted automatically to maintain a constant, but low density of activated fluorophores (Mund et al., 2018). The first microscope is equipped with a 60×/NA 1.49 TIRF objective (Nikon) and an iXON Ultra EMCCD camera (Andor). The second microscope is equipped with a 160×/1.43 NA objective (HCX PL APO, Leica) and an Evolve512D EMCCD camera (Photometrics). This microscope is additionally equipped with a laser speckle reducer for homogenous illumination (Deschamps et al., 2016).

We performed dual-color SMLM imaging with a 640-nm long pass dichroic mirror to split the emission signals from 640 and 561 nm laser excitation. The split signals were collected through 676/37 and 600/60 nm emission bandpass filters, respectively. Before imaging the cells, we first acquired images of 100-nm Tetra-Speck beads (catalog no. T7279, Thermo Fisher Scientific) for a faithful channel overlay. In brief, 0.75 µl Tetra-Speck beads were diluted in 360 µl $H_2O$, mixed with 40 µl 1 M $MgCl_2$, and put on a coverslip in a custom-manufactured sample holder. After 10 min, the mix was replaced with 400 µl $H_2O$. Images from different parts of the coverslips were collected with 50-ms exposure time. Next, when imaging the cells, ~120 mW of the 640 nm laser, ~70 mW of the 561 nm laser, 30 ms exposure time, and the self-adjusting UV laser were applied. An acquisition finished at maximally 60,000 frames.

For protein counting experiments, two strains expressing the Nup188-mMaple standard (Thevathasan et al., 2019) and the target kinetochore protein labeled with mMaple were mixed and imaged simultaneously. 225 regions were imaged per coverslip, separated by at least 150 µm to avoid premature mMaple activation. Every acquisition was performed with ~100 mW of the 561 nm laser, 25 ms exposure time, and the self-adjusting UV laser. All measurements were performed until all mMaple fluorophores had been activated and bleached. A snapshot of Ndc80-GFP (for kinetochores) or Abp1-GFP (for Nup188-mMaple strain) was automatically acquired, as well as a back focal plane image to exclude acquisitions with air bubbles.

### Single-molecule localization

We used the SMAP (Superresolution Microscopy Analysis Platform; Ries, 2020) for all data analysis. For single-molecule fitting, candidate localizations were detected by smoothing with a difference of Gaussians filter and thresholding. Then, the signal was localized by fitting a Gaussian function with a homogeneous photon background, treating the size of the Gaussian as a free-fitting parameter. Fluorophores spanning consecutive frames and thus likely stemming from the same fluorophore were merged (grouped) into a single localization. For experiments longer than 5,000 frames, cross-correlation-based sample drift correction was applied as described in Mund et al. (2018). Super-resolution images were reconstructed by rendering each localization as a Gaussian with a size proportional to the localization precision. Finally, localizations were filtered by localization precisions to exclude dim emitters and by PSF sizes to exclude out-of-focus fluorophores. If the localization density in the first frames was above the single-molecule regime, these frames were discarded.

Dual-color bead images were fitted as described above and used to calculate a projective transformation between the channels.

For high-throughput data, we extracted additional parameters for quality control such as the number of localizations and the median localization precision, photon count, PSF size, and background, and used them in combination with the BFP images to exclude poor measurements that resulted from air bubbles in the immersion oil or acidification of the buffer.

### Z-position bead calibration

A bead sample was prepared as described above. Next, using Micro-Manager (Edelstein et al., 2014), about 20 positions on the coverslip were defined and the beads were imaged acquiring z stacks (−1 to 1 µm, 10 nm step size) using the same filters as above. Images of beads were then localized to quantify their PSF sizes. Based on the PSF sizes and the stack positions, the z positions of fluorophores can be calibrated (Fig. S2 D).

### Quantification of distances between kinetochore proteins

We quantified distances between kinetochore proteins based on a cross-correlation analysis. Before the analysis, in a dual-color SMLM data set, localizations with localization precision >20 nm for Alexa Fluor 647 and >25 nm for mMaple channels or PSF size <100 or >160 nm were removed. Only the in-focus structures

(mean PSF size ≤135 nm) were kept for the analysis. One color/channel (usually the channel of Spc105 unless specified otherwise) was defined as the reference and the other as the target. We started by manually collecting kinetochore clusters (sites) and grouped both kinetochore clusters of the same mitotic spindle as a pair (Fig. S1 A). For each pair, a line was manually drawn to represent the spindle axis, which the kinetochore clusters distributed along. Next, to take the opposite direction of chromosomes pulling by each kinetochore cluster of the pair into account, the axial direction was defined as pointing toward the center of the spindle (Fig. S1 A). As shown in Fig. S1 B, each kinetochore cluster/pair of kinetochore clusters went through the same analysis steps (Fig. S1, C and D) for quantifying the distance. First, we calculated the image cross-correlation between two reconstructed super-resolution images corresponding to the two channels for each kinetochore cluster separately. From the maximum position of the cross-correlation map, we determined the average distance between the two proteins along the spindle axis. To eliminate the effect of residual transformation errors, caused, e.g., by chromatic aberrations, we always analyzed the two paired kinetochore clusters together. Due to their close proximity, we expect similar registration errors, which cancel out when calculating the average protein distance because of the opposite orientation of the kinetochore clusters. As a result, each spindle resulted in one average distance value. Using Spc105 as a reference in most data sets, we could position all measured proteins along the spindle axis. The number of experiments per kinetochore protein is summarized in Tables 1 and S4.

**Estimation of the error introduced by axial tilts of spindle axes**
We first quantified the average width of kinetochore clusters based on a cylindrical distribution. Specifically, the 1D profile along the diameter of a cylinder convolved with a Gaussian function ($\sigma$ defined as the mean localization precision) was calculated. Such a profile was fitted to kinetochore clusters with the radius as a free parameter.

We localized emitters in the bead $z$-stacks acquired as described above to obtain their PSF sizes. We then fitted a quadratic curve to the scatter plot of the PSF sizes and $z$ positions of beads. The fitted calibration curve describes the relation between $z$ positions of localizations and PSF size.

The 1D profile of cylindrical distribution with the radius defined as the quantified average width of kinetochore clusters was plugged into the calibration curve to obtain a new calibration curve describing the relation between $z$ position of a kinetochore cluster and its mean PSF size. We then drew a line at mean PSF size = 135 nm, which is the maximal possible value of the analyzed kinetochore clusters (Fig. S2 E). The maximal axial distance between kinetochore clusters in the same pair $d_z^{max}$ is defined as the distance between the crosspoints of the line and the calibration curve. The distance between the two kinetochore clusters in 3D was estimated as $d = \sqrt{d_{xy}^2 + d_z^2}$, where $d_{xy}$ is the lateral distance between the two kinetochore clusters. The relative error introduced by the axial tilt is calculated as $\epsilon(\theta) = (d - d_{xy})/d$, where $\theta = \cos^{-1}(d_{xy}/d)$ is the tilt angle. The maximum tilt angle $\theta^{max}$ was estimated based on the mean lateral distance $\overline{d_{xy}}$ and the estimated maximum axial distance $d_z^{max}$. The mean error is then estimated as $\overline{\epsilon} = (\int_0^{\theta=\theta^{max}} \epsilon(\theta))/\theta^{max}$.

**Estimations of the widths of kinetochore protein distributions**
We used autocorrelation analysis to quantify the widths of kinetochore protein distributions. For each kinetochore cluster, we generated a 2D autocorrelation map with a pixel size of 5 nm. Each map was further converted to a 1D profile by summing the autocorrelation values within 25 nm shifts along the spindle. The resulting profile represents the sum autocorrelation across the shift perpendicular to the spindle axis. The high sum autocorrelation value at shift = 0 was substituted by the value of its neighboring shift. The profile was then normalized to have a maximum of 1 before averaging over all kinetochore clusters of the same kinetochore proteins. To separate the real autocorrelation from its background, two Gaussian functions with a linked parameter $\mu$ (position) were then fitted to the averaged profile. The function with the larger fitted parameter $\sigma$ was considered as the background and then subtracted from the averaged profile. This profile for each analyzed protein is shown Fig. S4.

We performed simulations to obtain reference autocorrelation profiles of ring distributions with different radii. Specifically, the 1D profile along the diameter of a ring was calculated per specified radius. To take the experimental localization precision into account, we acquired its binned distribution based on the mMaple channel over all the dual-color data sets. We then convolved the 1D profile with a Gaussian function ($\sigma$ taken from the bin value) per bin. We then summed the profiles weighed by the frequency of the corresponding bins to form the final profiles. For each final profile, its autocorrelation was then calculated and is shown in Fig. S4.

**Protein copy number estimations**
To differentiate the yeast strains (counting reference standard or kinetochore proteins) on the same coverslip, proteins with different cellular distributions were tagged with mEGFP in the reference and target strains (Abp1 for the reference and a kinetochore protein for the target). The GFP signal was checked in the diffraction-limited channel. We then manually segmented the single structures of the reference (NPCs) and the target (kinetochore clusters) in respective strains. Before further analysis, localizations with localization precision >15 nm or PSF size <100 or >170 nm were removed. Only the in-focus structures (mean PSF size ≤135 nm) were retained in the analysis. NPCs at the edge of the nucleus or too close to neighboring structures were excluded. We then determined the number of localizations in a circular region of interest of a diameter of 150 nm. For a target structure, we only picked kinetochore clusters that have two foci in the GFP channel to ensure metaphase kinetochore clusters. We then determined the number of localizations in the manually created polygon enclosing the kinetochore cluster. When paired kinetochore clusters were too close to each other, they were segmented as one entity and their localizations were divided by 2. The copy number calibration factor for each dataset was calculated as $F_n = L_n/N_n$, based on the stoichiometry of Nup188 (Table S5). Here, $L_n$ is the mean

**Table 1. Statistics of kinetochore protein positions along the spindle axis**

| Protein | Distance (nm) | SD | SEM | N |
|---|---|---|---|---|
| | to Spc105 | | | |
| Ask1 | 44.3 | 9.3 | 2.4 | 15 |
| Ndc80 | 13.6 | 6.1 | 1.2 | 25 |
| Nuf2* | 16.9 | 3.3 | 1.5 | 5 |
| Spc25 | −2.5 | 2.9 | 0.8 | 13 |
| Nsl1 | 6.5 | 4.3 | 1.5 | 8 |
| Nnf1 | 4.8 | 6.9 | 2.6 | 7 |
| Mtw1 | 4.3 | 2.5 | 0.6 | 17 |
| Dsn1 | 3.1 | 2.3 | 0.6 | 13 |
| Chl4 | −23.5 | 11.5 | 2.9 | 16 |
| Ctf19 | −14.9 | 4.7 | 1.7 | 8 |
| Okp1 | −13.4 | 4.8 | 1.4 | 12 |
| Cnn1 | −20.1 | 7.6 | 2.7 | 8 |
| Mif2 | −23.8 | 6.3 | 2.0 | 10 |
| Cep3 | −21.1 | 4.2 | 1.7 | 6 |
| Cse4 | −16.9 | 4.2 | 1.3 | 10 |
| | to Ndc80 | | | |
| Ctf19 | −24.9 | 8.5 | 1.8 | 23 |

*The position of Nuf2 is based on the measured pair Ndc80–Nuf2. N: number of spindles.

quantified localizations per NPC and $N_n$ = 16 is the known copy number of Nup188 per NPC. Then the mean copy number $N_k$ of a target protein per kinetochore was calculated as $N_k = \frac{\left(\frac{L_{kc}}{N_{kc}}\right)}{F_n}$, where $N_{kc}$ = 16 is the number of kinetochores per kinetochore cluster and $L_{kc}$ is the mean quantified localizations per kinetochore cluster. To take the variation of the NPC localizations into account, the standard deviation of the kinetochore protein copy number was $S = N_k \sqrt{(S_n/M_n)^2 + (S_k/M_k)^2}$, where $S_k$ and $S_n$ are the standard deviations of the localization counts for NPC and kinetochore protein. $M_k$ and $M_n$ are the respective sample sizes. In this work, ∼600 NPCs ($M_n$ = ∼600) were analyzed for each replicate. Finally, for each target kinetochore protein, the pooled copy number and standard deviation of replicates were given by $\bar{N}_k = \sum_i (N_{ki}M_{ki})/\sum_i M_{ki}$ and $\bar{S} = \sqrt{\sum_i ((M_{ki} - 1)S_i^2)/\left(\sum_i M_{ki} - I\right)}$, respectively. Here, $M_{ki}$, $N_{ki}$, and $S_i$ are the sample size, quantified kinetochore copy number, and its SD of the $i^{th}$ of total $I$ replicates ($I$ = 2 in this work), respectively. The pooled SEM was calculated as $\bar{S}_m = \bar{S}/\sqrt{\sum_i M_{ki}}$.

**Online supplemental material**

Fig. S1 shows the workflow of quantifying the distances between kinetochore proteins. Fig. S2 shows the basis for defining the values for filtering and quality control. Fig. S3 shows intra-kinetochore distances measured by different approaches. Fig. S4 shows autocorrelation perpendicular to the spindle axis. Fig. S5 shows protein copy numbers per kinetochore measured with different filtering or treatments. Table S1 shows the comparison of the available distance measurements from this article and Joglekar et al. (2009). Table S2 shows the yeast strains created and used in this study. Table S3 shows the PCR primers used in this study. Table S4 shows additional information about the dual-color SMLM experiments. Table S5 shows the calibration factors for protein counting.

## Acknowledgments

We thank Andrea Musacchio and Ulrike Endesfelder for feedback on the manuscript, and Katharina Lindner for her work on the imaging of the dual-color strains.

This work was supported by the European Research Council (grant no. ERC CoG-724489 to J. Ries), the Human Frontier Science Program (RGY0065/2017 to J. Ries), and the European Molecular Biology Laboratory.

Author contributions: K. Cieslinski and J. Ries conceived the study. K. Cieslinski, Y.-L. Wu, and J. Ries managed the project. K. Cieslinski, Y.-L. Wu, S.J. Hörner, and J. Ries developed the methodology. K. Cieslinski, S.J. Hörner, Y.-L. Wu, and M. Skruzny performed experiments. J. Ries and Y.-L. Wu developed data analysis. K. Cieslinski, Y.-L. Wu, L. Nechyporenko, M. Skruzny, and J. Ries analyzed data. K. Cieslinski, Y.-L. Wu, and L. Nechyporenko curated data. K. Cieslinski and Y.-L. Wu validated the results. K. Cieslinski, Y.-L. Wu, and L. Nechyporenko visualized the results. J. Ries acquired funding and supervised the

study. K. Cieslinski, Y.-L. Wu, D. Conti, and J. Ries wrote the manuscript with input from all authors. All authors reviewed and edited the manuscript.

Disclosures: The authors declare no competing interests exist.

Submitted: 23 September 2022

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

# Supplemental material

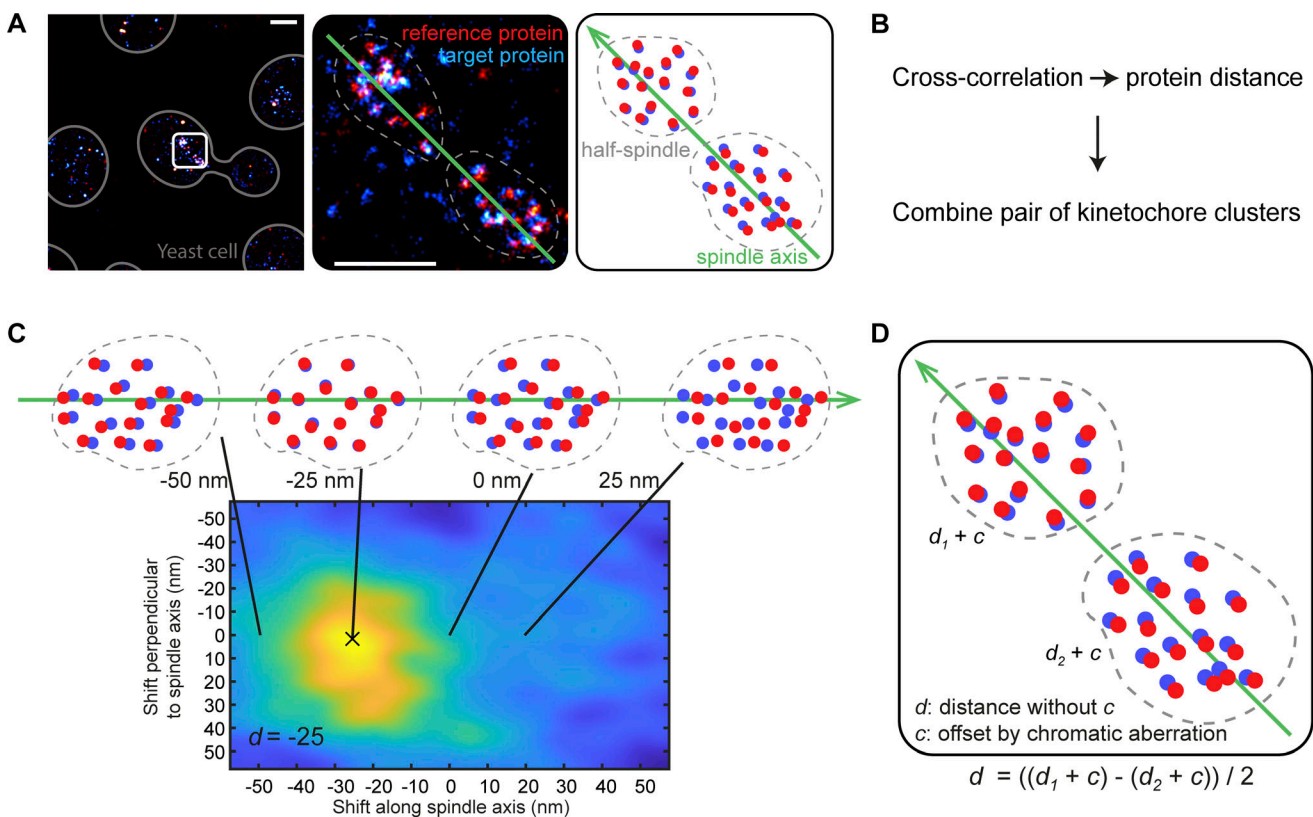

Figure S1. **Workflow of quantifying the distances between kinetochore proteins. (A)** Metaphase spindles (white box) with both half spindles close to the focus are manually segmented (dashed contour). The spindle axis for each spindle is manually annotated (green line). A schematic (right panel) is provided for clarity. Scale bars: 1 μm (left), and 500 nm (middle). **(B)** The overview of the workflow. **(C)** The distance between the target and reference proteins is quantified using the cross-correlation analysis. This analysis is applied to each kinetochore cluster and yields a correlation map showing the similarity between the two channels at certain lateral and axial shifts of the reference channel. The shift along spindle axis at the maximum is quantified as the distance $d$. **(D)** To eliminate the potential offset $c$ caused by the chromatic aberration, the average distances $d$ of both paired kinetochore clusters, having the distances $d_1$ and $d_2$ respectively, is then calculated per spindle.

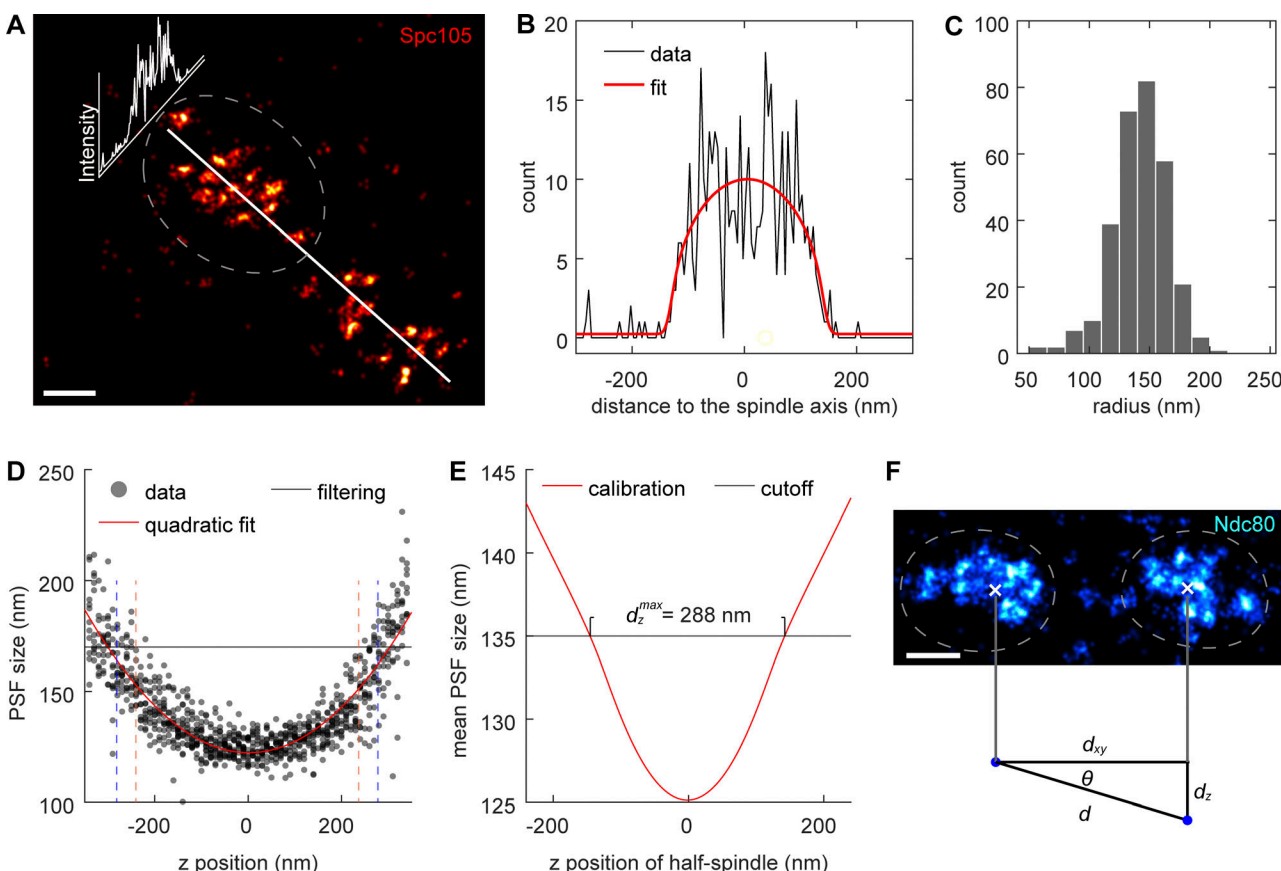

Figure S2. **The basis for defining the values for filtering and quality control. (A–C)** Quantifying the width of kinetochore clusters. As shown with the example kinetochore cluster (A), its profile perpendicular to the axis of spindle (B) was fitted with a cylindrical model (red) to quantify the radius. **(C)** The radius of analyzed kinetochore clusters. The mean radius was quantified as 142.0 ± 23.7 (SD) nm, which corresponds to the width (diameter) of 284 nm. Sample size: 301 kinetochore clusters. **(D)** The calibration curve (red) relating $z$ positions to PSF size based on bead data (dots). For filtering out out-of-focus localizations, the maximum PSF size of 170 nm is defined, which corresponds to an axial range from −300 to 300 nm. The z ranges bounded by the vertical dashed lines with the same colors [mean PSF size cutoff: 130 nm (orange), 135 nm (blue)] are where kinetochore proteins can be found, given the corresponding mean PSF size cutoffs of kinetochore clusters, taking the quantified width in C into account. Both cutoffs ensure that no analyzed kinetochore protein exceeds the imaging depth determined by the PSF size filtering. **(E)** The calibration curve relating the $z$ position of a kinetochore cluster to its mean PSF size based on the bead calibration in D. The maximal axial distance between kinetochore clusters in the same pairs $d_z^{max}$ is estimated to be 288 nm, given that the maximal allowed mean PSF size is 135 nm. **(F)** The relation between the lateral distance $d_{xy}$, the axial distance $d_z$, and the estimated distance between kinetochore clusters in the same pairs $d$ in 3D. Based on the dataset (Ndc80) with the largest sample size, the mean lateral distance between kinetochore clusters in the same pairs $\bar{d}_{xy}$ is measured as 777 nm. These correspond to the maximum tilt angle $\theta^{max}$ = 20.3°, maximum tilt-introduced error of the distance between the kinetochore clusters $\epsilon^{max}$ = 6.3%, and the mean error $\bar{\epsilon}$ = 2.1%. See Materials and methods for the calculations. Sample size: 50 kinetochore clusters. Scale bars: 200 nm.

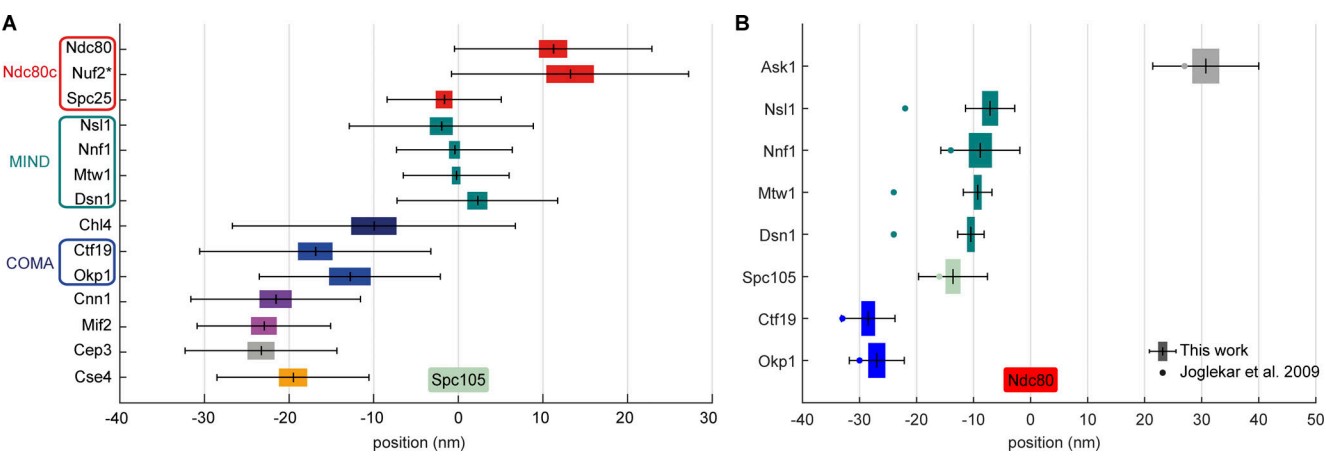

Figure S3. **Intrakinetochore distances measured by different approaches. (A)** An independent analysis of intrakinetochore distances based on manually picked single kinetochores. The mean distance is plotted with SEM (as colored box) and SD (whiskers). *The position of Nuf2 was estimated based on Nuf2–Ndc80 distance measurements. **(B)** Comparison of the available distance measurements to Joglekar et al. (2009). The mean distance is plotted with SEM (as colored box) and SD (whiskers).The corresponding mean values reported by Joglekar et al. (2009) are shown as dots. For comparison, our distance measurements were recalculated using the Ndc80 as the reference point.

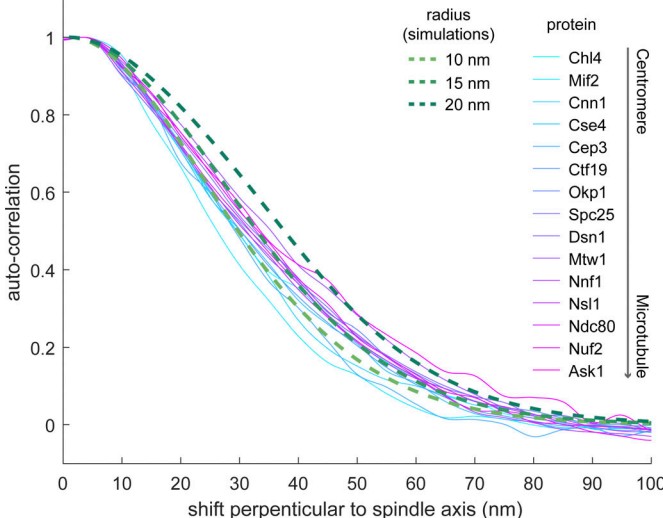

Figure S4. **Autocorrelation perpendicular to the spindle axis.** Solid curves are average autocorrelation profiles of kinetochore proteins. Dashed lines are autocorrelation profiles of simulated ring distributions with corresponding radii considering the overall distribution of the experimental localization precision.

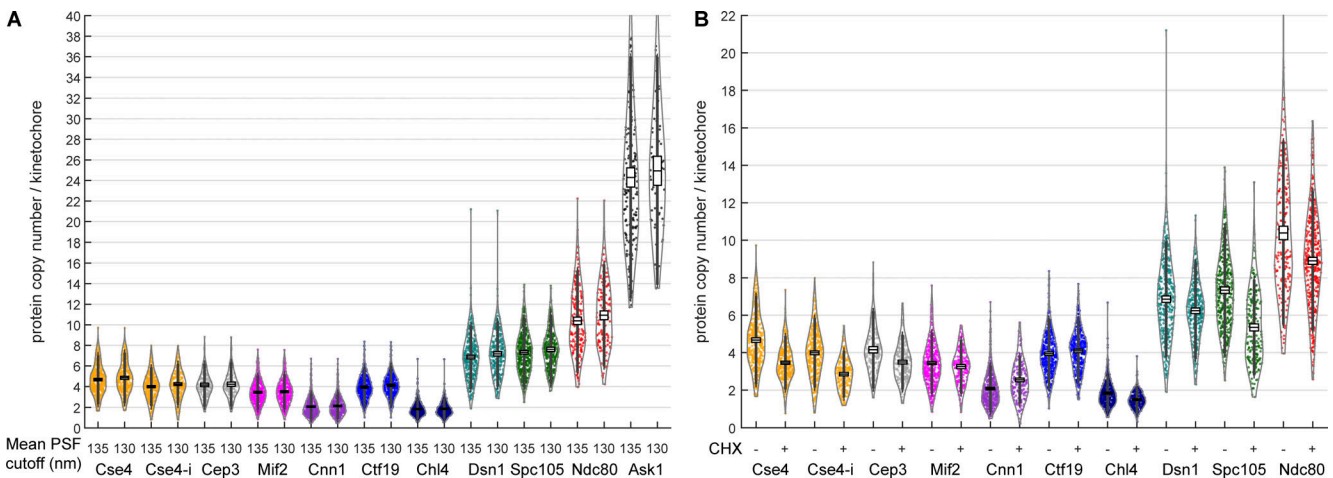

Figure S5. **Protein copy numbers per kinetochore measured with different filtering or treatments. (A)** To investigate the robustness of the molecular counting, different filtering by mean PSF size of kinetochore clusters were applied. Either the kinetochore clusters with PSF size ≤135 or 130 nm were analyzed. The mean protein copy numbers calculated based on both cutoffs are almost identical, showing that the analysis is robust. **(B)** Cells were treated with or without CHX (250 µg/ml, 60 min) to investigate the effect of protein maturation. Each data point corresponds to one kinetochore cluster. Boxes denote average copy numbers and SEMs, and whiskers denote SDs.

**Provided online are five tables. Table S1 shows the comparison of the available distance measurements from this article and Joglekar et al. (2009). Table S2 shows the yeast strains created and used in this study. Table S3 shows the PCR primers used in this study. Table S4 shows additional information about the dual-color SMLM experiments. Table S5 shows calibration factors for protein counting.**

