## [Peer Review File · The Journal of Cell Biology]

Nanoscale structural organization and stoichiometry of the budding yeast kinetochore

Konstanty Cieslinski, Yu-Le Wu, Lisa Nechyporenko, Sarah Hörner, Duccio Conti, Michal Skruzny, and Jonas Ries

Corresponding Author(s): Jonas Ries, European Molecular Biology Laboratory

Review Timeline:

Submission Date:	2022-09-23
Editorial Decision:	2022-11-21
Revision Received:	2022-12-16

Monitoring Editor: Arshad Desai

Scientific Editor: Dan Simon

Transaction Report:

DOI: <https://doi.org/10.1083/jcb.202209094>

Revision 0

Review #1

1. Evidence, reproducibility and clarity:

Evidence, reproducibility and clarity (Required)

The authors have developed a rigorous methodology for using single-molecule imaging of exogenously labeled kinetochore proteins to count and estimate their copy numbers and the average distance from the kinetochore protein Spc105. Although the method is technically sound, its application to the kinetochore raises some crucial questions below. My biggest concern is the effect of non-centromeric pools of the centromeric proteins Cse4, Cep3, and Ctf19 on the estimated copy number per kinetochore. The authors should be able to address most, if not all, questions by presenting a more in-depth data analysis.

1. Accounting for tilt of the yeast spindle relative to the image plane: It is not clear to me how the authors ascertain whether the spindle being imaged is nearly parallel to the image plane. In the companion fission yeast study, spindle poles are used for this purpose, but this study seems to rely only on the labeled kinetochore proteins. The criteria used to select the in-plane spindles should be clearly defined.

2. The effects of PSF depth on counting kinetochore proteins: The authors use a well-characterized nuclear pore protein as the reference to estimate kinetochore protein counts per half-spindle. Although this method appears rigorous in principle, I am unsure about the effect of the spatial distribution of kinetochores on the accuracy of the estimated number. Nuclear pore proteins are all localized within an $< 100 \text{ nm}^3$ volume. Therefore, all proteins within an in-focus nuclear pore will also be in focus. This is not the case with yeast kinetochores, especially in metaphase. A fraction of the kinetochores is likely to be $> 100 \text{ nm}$ away from the focal plane even when the spindle is perfectly parallel to the focal plane. A discussion of this possibility, its effect on the protein count/distance estimates, and any mitigating factors is essential to highlight the caveats associated with the conclusions.

3. Presentation of the cross-correlation analysis: The authors use cross-correlation for an unbiased calculation of the axial separation between a protein of interest and Cse4, but I am curious about the structure of the underlying data, and the intensity image in Figure 1 is not easy to examine. It will be helpful to include more analysis of the underlying data for at least a subset of the proteins (e.g., proteins at short, intermediate, and long distances from Cse4) as supplementary data.

- The authors should include X and Y projections of the cross-correlation function.
- Do the widths of cross-correlation functions (i.e., their spread perpendicular to the spindle axis) match across all proteins and experiments? This should be an almost invariant characteristic of the measurements, assuming that proteins within each kinetochore tightly cluster around the 25 nm microtubule. This line of thinking makes the large width of the cross-correlation shown in Figure 1 somewhat surprising.
- It will also be interesting to test if the correlation between the positions of Spc105 molecules,

especially perpendicular to the spindle axis, is comparable to the known separations between adjacent microtubules in the yeast spindle (the authors could use Winey et al. 1995 for serial-section EM of yeast spindles for comparison).

4. Cse4 count (4 per kinetochore) and the model presented: One of the surprising conclusions of the study is that there are two nucleosomes associated with each microtubule attachment, with Mif2/CENP-C potentially interacting with both nucleosomes. There are two critical issues that the authors must consider.

- (1) Fluorescent protein chimeras of Cse4 and CBF3 and COMA complex members do not exclusively localize to kinetochores. Biochemical studies show that both Cse4 and CBF3 proteins interact with non-centromeric DNA, e.g., see work from the Biggins lab regarding Cse4 over-expression and also from the Henikoff group that used ChIP-seq. I can't think of a similar reference for the CBF3 complex, but the DNA-binding proteins are also likely to interact with other parts of the genome. The non-centromeric protein is visible as a significant background fluorescence in wide-field microscopy, e.g., see Cep3 localization here:

<https://images.yeastrc.org/imagerepo/viewExperiment.do?id=202308&experimentGroupOffset=3&experimentOffset=0&experimentGroupSize=3>

Similar background fluorescence can be detected for Cse4 and Ctf19. This extra-centromeric localization of Cse4, Cep3, and Ctf19 makes it possible that the protein counts included by the authors are "contaminated" to some extent by the extra-centromeric protein. The authors should discuss this possibility and how it might affect their counts.

- (2) The model drawn in Figure 4 makes explicit assumptions about the positioning of the four Cse4 molecules (or two nucleosomes) in each kinetochore relative to the rest of the kinetochore components. Yet, the data shown do not justify this specific arrangement. Lawrimore et al. 2011 claim that the non-centromeric Cse4 nucleosomes must be randomly distributed in the pericentromeric chromatin to evade detection in biochemical tests. Therefore, the nearest-neighbor analysis suggested above will be valuable for gaining new insights into the relative positioning of the centromeric- and non-centromeric Cse4 nucleosomes. A similar analysis for Cep3 and Ctf19 will also be helpful. If stereotypical positioning of these molecules cannot be detected, then the model should be revised accordingly (alternative models that are also consistent with the data can be included).

- (3) I suggest one experiment that can help the authors better understand protein organization in one kinetochore. Joglekar et al. 2006 used a dicentric chromosome to isolate single kinetochores on the spindle axis to test the assumption that each kinetochore consists of approximately the same number of molecules of kinetochore proteins. The strains are easy to construct (transform existing strains with a linearized plasmid). Single kinetochores can be seen with a low but reasonable frequency. I leave the decision to perform the experiment to the authors' discretion depending on whether the experiment will be worth the effort in strengthening or enhancing their conclusions.

5. Information regarding the degree of correction applied to calculate protein count per half-spindle: It will be helpful to include data regarding the degree of correction applied to the expected and measured numbers of NPC protein as supplementary data so that the readers can see the magnitude of this correction relative to the measured counts.

****Minor points:****

1. McIntosh et al. JCB 2013 used microtubule plus-ends in serial section electron micrographs of

yeast spindles to align the centromeric region and found a disk-shaped structure that roughly corresponds to the size of a single nucleosome ~ 80 nm away from the tip of the microtubule and centered the microtubule axis. The authors should refer to this finding in their discussion of the model that they present with two nucleosomes. In my opinion, this is compelling evidence for a nucleosome-like structure serving as the kinetochore foundation.

2. As discussed by the authors, the number of Cse4 molecules per kinetochore has been the subject of some controversy. Biochemical data from the Biggins group and ChIPseq data from the Westermann group (Altunkaya et al. 2016 Current Biology) strongly suggest that Cse4 molecules can only be found centered on the centromeric sequence. The latter reference should be included in the discussion.

3. Although microscopy-based methods have estimated anywhere from 1, 2, to 6 Cse4 molecules per kinetochore, these studies generally agree on the stoichiometry between Cse4 and the rest of the kinetochore proteins, e.g., Ndc80 complex proteins are ~ 4-fold more abundant than Cse4, etc. The present study seems to disagree with protein stoichiometry. The authors may find it worthwhile to note this feature of their data.

4. Omission of the Dam1 complex from this study is disappointing to me personally, but I am sure that the authors have good reasons for this. They should briefly comment on the absence of the Dam1 complex in this study.

2. Significance:

Significance (Required)

Cieslinski and colleagues present a single-molecule localization-based study to define the copy numbers and relative organization of kinetochore proteins in budding yeast. These numbers confirm and significantly refine prior measurements of the same aspects of the kinetochore. They also raise new questions and point to new research directions. The measurements also reveal a model of the protein organization of the budding yeast kinetochore in metaphase. For these reasons, the manuscript is of significant interest to the cell division field.

3. How much time do you estimate the authors will need to complete the suggested revisions:

Estimated time to Complete Revisions (Required)

(Decision Recommendation)

Between 1 and 3 months

Review #2

1. Evidence, reproducibility and clarity:

Evidence, reproducibility and clarity (Required)

In this study, Cielinski and colleagues have applied single molecule localization microscopy to map the positions of proteins in the yeast kinetochore. This has not been reported previously and this study is both well-conducted and the data appear solid. They also use a modification of this technique to assess the stoichiometry of kinetochore proteins. The results that they obtain are broadly in line with several previous studies that use other methodology. There may be an improvement in accuracy using this new approach that has not been obtained previously and there are some important novel conclusions from this work. I would like the authors to address the following concerns prior to publication:

****Major points****

1. One interesting finding is that there is a discrepancy in the length of both the MIND and NDC80 complexes (from crystallographic data) with their relative positions. The authors suggest that the outer complexes could be twisted or rotated in respect of the spindle axis. It would be great if the authors could illustrate this in their model (or discuss it in the text), to demonstrate the required angle of twist/rotation of both complexes to account for the discrepancy. A twisted filament structure to the outer kinetochore does have some implications for its response to tension - a key determinant of kinetochore-microtubule attachment. It also may provide some flexibility to the structure under tension.
2. For the experiment with cycloheximide, the authors state "Although we observed minor changes in copy numbers, the overall effect of CHX was small." For some proteins, Cse4i for example, there appears to be a significant decrease in intensity (30-40%) after cycloheximide treatment, see Figure S3. While the conclusion that tag maturation does not affect copy number measurements is sound, I suggest modifying this section to reflect the data.
3. Page 5. The statement "These data agree reasonably well with previous diffraction-limited dual-color microscopy studies ..." provides readers with little ability to compare the data. I would like to see a supplementary figure comparing these new data with previous studies, especially those of Joglekar et al 2009, see Figure 3 in this paper.
4. In terms of the distances quoted, are they in one dimension (as per Jogelkar et al 2009) or in three? The results section is entitled "...positions of kinetochore proteins along the metaphase spindle axis", which suggests a single dimension. Please make this very clear in the results section. In the discussion, is the statement "we mapped the relative positions of 15 kinetochore proteins along the kinetochore axis", which is not entirely clear. It seems from the methods that this is one dimension "...we determined the average distance between the two proteins along the spindle axis." I suggest clarifying the results section briefly and clearly to indicate that this is a single dimension being measured and also using consistent wording of the axis measured throughout the text.

****Minor points:****

Abstract: I would drop "all" from "For all major kinetochore proteins...", since full characterisation was performed on 14 proteins (9 in terms of copy number).

Page 2: "trough" to through.

Page 2 "S. cerevisiae" to italics

Methods p11. How do the MKY strains relate to common yeast genetic backgrounds? (e.g. are they S288C?).

2. Significance:

Significance (Required)

This manuscript, together with an accompanying one from Virat et al., are nice complementary studies that provide the first single molecule localization studies of the yeast kinetochore. Although other labs have used super-resolution methods to study individual kinetochore proteins; both of these new studies map distances between many proteins at the kinetochore and thus are able to produce maps of the overall kinetochore structure. Like the previous study using standard resolution methods (Joglekar et al, 2009. Current Biology 19, 694-699); these studies will likely provide a benchmark for future studies on eukaryotic kinetochore architecture, including those in mammalian systems. Additionally, this work will appeal to super-resolution microscopists.

My expertise is as a yeast kinetochore cell biologist.

3. How much time do you estimate the authors will need to complete the suggested revisions:

Estimated time to Complete Revisions (Required)

(Decision Recommendation)

Between 1 and 3 months

Manuscript number: RC-2021-01179

Corresponding author(s): Jonas Ries

1. General Statements [optional]

In our work, we quantified the abundance and positions of major kinetochore proteins within the metaphase kinetochore in budding yeast using single-molecule localization microscopy. Based on these measures, we revised the current model of the kinetochore and provided a nanoscale view of the complex.

We now revised our manuscript according to reviewers' points. We performed new analyses to quantify the measurement errors and to justify our data analysis workflows. We further exploited the correlation-based analysis and found a correlation between the spreads of kinetochore proteins perpendicular to the spindle axis and their positions along the axis. We also discussed the potential non-centromeric pools and revised our model of the kinetochore. Further information on our analyses was now provided to improve the clarity. Changes to the text were implemented to better reflect our data. Information from relevant works was incorporated to better connect this work to the field.

We thank the reviewers for their points, which help us show the rigorosity of our analyses, further demonstrate the potential of our work, and improve clarity.

Reviewer #1 (Evidence, reproducibility and clarity (Required)):

The authors have developed a rigorous methodology for using single-molecule imaging of exogenously labeled kinetochore proteins to count and estimate their copy numbers and the average distance from the kinetochore protein Spc105. Although the method is technically sound, its application to the kinetochore raises some crucial questions below. My biggest concern is the effect of non-centromeric pools of the centromeric proteins Cse4, Cep3, and Ctf19 on the estimated copy number per kinetochore. The authors should be able to address most, if not all, questions by presenting a more in-depth data analysis.

Major points

1. Accounting for tilt of the yeast spindle relative to the image plane: It is not clear to me how the authors ascertain whether the spindle being imaged is nearly parallel to the image plane. In the companion fission yeast study, spindle poles are used for this purpose, but this study seems to rely only on the labeled kinetochore proteins. The criteria used to select the in-plane spindles should be clearly defined.

We thank the reviewer for pointing this out. We selected the in-plane spindles based on their average PSF size, which informs the z positions of the center of the kinetochore cluster (for simplicity, now all 'half-spindle' was changed to 'kinetochore cluster'). To calibrate the z position of kinetochore clusters, we first measured the width of the kinetochore cluster by fitting a cylindrical distribution. Overall, the kinetochores are likely symmetrically distributed around the spindle axes. Therefore, the height and the width of a kinetochore cluster should be the same. We then calibrated the z positions

of the PSF size based on fluorescent bead data. Next, we plugged in the cylindrical distribution to the calibration curve to correlate the mean PSF size and z position of the kinetochore cluster. We only took the kinetochore clusters with a mean PSF size < 135 nm, which corresponds to ~144 nm away from the focal plane. In the worst case when the paired kinetochore clusters are on different sides of the focal plane, their distance in z is 288 nm. This distance corresponds to a tilt angle of 20.3° and a maximum error of 6.3%, given the mean measured lateral distance of 777 nm. Assuming a random tilt angle from 0° to the maximum, we calculated the average error as 2.1%. Therefore, the effect of the tilt on the quantification can be ignored. These results are now included in a new **Fig. S1** and described on **page 5 lines 130-131** and **page 14 lines 481-499**. For consistency, we now applied the unified filtering (**page 14 lines 460-463**) to all analyzed datasets and updated all the distance measures as well as **Fig. 2, Fig. 4** and **Table 1**.

2. The effects of PSF depth on counting kinetochore proteins: The authors use a well-characterized nuclear pore protein as the reference to estimate kinetochore protein counts per half-spindle. Although this method appears rigorous in principle, I am unsure about the effect of the spatial distribution of kinetochores on the accuracy of the estimated number. Nuclear pore proteins are all localized within an < 100 nm³ volume. Therefore, all proteins within an in-focus nuclear pore will also be in focus. This is not the case with yeast kinetochores, especially in metaphase. A fraction of the kinetochores is likely to be > 100 nm away from the focal plane even when the spindle is perfectly parallel to the focal plane. A discussion of this possibility, its effect on the protein count/distance estimates, and any mitigating factors is essential to highlight the caveats associated with the conclusions.

Based on the cylindrical distribution (see please the reply to point 1) of kinetochore clusters and their positions in z, we calculated the upper and lower boundaries of the distribution of kinetochore proteins in z, given a specific mean PSF size cutoff of a kinetochore cluster. Regardless of how stringent the cutoff is (130 and 135 nm), we made sure the boundaries do not exceed the imaging depth defined by our choice of the PSF size filtering (<170 nm). This was confirmed by the consistent measured copy numbers across the two different cutoffs, as shown in the new **Fig. S6**. This point is now explained on **page 7 lines 190-192**. For consistency, we now applied the unified filtering (**page 15 lines 522-524**) to all analyzed datasets and updated all the copy number measures as well as **Fig. 3, Fig. 4, Fig. S7**, and **Table S3**.

3. Presentation of the cross-correlation analysis: The authors use cross-correlation for an unbiased calculation of the axial separation between a protein of interest and Cse4, but I am curious about the structure of the underlying data, and the intensity image in Figure 1 is not easy to examine. It will be helpful to include more analysis of the underlying data for at least a subset of the proteins (e.g., proteins at short, intermediate, and long distances from Cse4) as supplementary data.

- The authors should include X and Y projections of the cross-correlation function.
- Do the widths of cross-correlation functions (i.e., their spread perpendicular to the spindle axis) match across all proteins and experiments? This should be an almost invariant characteristic of the measurements, assuming that proteins within each kinetochore tightly cluster around the 25 nm microtubule. This line of thinking makes the large width of the cross-correlation shown in Figure 1 somewhat surprising.
- It will also be interesting to test if the correlation between the positions of Spc105 molecules, especially perpendicular to the spindle axis, is comparable to the known separations between

adjacent microtubules in the yeast spindle (the authors could use Winey et al. 1995 for serial-section EM of yeast spindles for comparison).

The reviewer is interested in the spread, or the size of the distribution, of a protein in a kinetochore along and perpendicular to the spindle axis. This is an interesting idea and can be done practically. However, the information can be more easily obtained based on auto-correlation instead of cross-correlation, due to its better signal-to-noise ratio along the dimension perpendicular to the spindle axis. Cross-correlations in that dimension are convoluted with background localizations and different localization precisions of the two channels. These factors are hard to interpret and disentangled. In auto-correlations, although the background is still present, it can be modeled and then removed easily, as now mentioned on **page 15 lines 500-516**.

Accordingly, we performed auto-correlation analysis on all the proteins and compared them to simulations representing different sizes. We find that the size of the distribution correlates to the position of the protein along the spindle axis. The results are now included as the new **Fig. S5** and discussed on **page 6 lines 169-176**.

The cross-correlation analysis was based on only the position of the maximum value, not the projections. To keep the figure concise, we decided not to include the projections. However, the auto-correlation analysis was indeed based on projections, which we now included in **Fig. S5**.

Regarding the correlation between the positions of Spc105 molecules, we believe the reviewer actually refers to the correlation between the positions of kinetochores. Auto-/cross-correlations contain the information of the cluster sizes, based on the first peak (as shown in **Fig. S5**), and the relative distance (if the pattern is periodic). Unfortunately, the positions of kinetochores perpendicular to the spindle axis are not periodically distributed. Therefore, we cannot comment on the separations between adjacent microtubules.

4. Cse4 count (4 per kinetochore) and the model presented: One of the surprising conclusions of the study is that there are two nucleosomes associated with each microtubule attachment, with Mif2/CENP-C potentially interacting with both nucleosomes. There are two critical issues that the authors must consider.

(1) Fluorescent protein chimeras of Cse4 and CBF3 and COMA complex members do not exclusively localize to kinetochores. Biochemical studies show that both Cse4 and CBF3 proteins interact with non-centromeric DNA, e.g., see work from the Biggins lab regarding Cse4 over-expression and also from the Henikoff group that used ChIP-seq. I can't think of a similar reference for the CBF3 complex, but the DNA-binding proteins are also likely to interact with other parts of the genome. The non-centromeric protein is visible as a significant background fluorescence in wide-field microscopy, e.g., see Cep3 localization here: <https://images.yeastrc.org/imagerepo/viewExperiment.do?id=202308&experimentGroupOffset=3&experimentOffset=0&experimentGroupSize=3>

Similar background fluorescence can be detected for Cse4 and Ctf19. This extra-centromeric localization of Cse4, Cep3, and Ctf19 makes it possible that the protein counts included by the authors are "contaminated" to some extent by the extra-centromeric protein. The authors should discuss this possibility and how it might affect their counts.

After consideration, we agree with the reviewer that, specifically, a fraction of counted Cse4 molecules should be considered non-centromeric. We agree that the previous data is certainly sufficient to conclude it. The reviewer made a similar suggestion about COMA and CBF3 subcomplexes. In recent

years a substantial portion of inner kinetochore components has been reconstituted. In Harrison et al. 2019, the Ctf19 complex structure has been solved. Two copies of the complex were observed. Therefore, the non-centromeric pool of COMA is certainly possible and we now made the adjustments to the text (**page 8, lines 219-225**) and **Fig. 4**. Accordingly, we now also modified the abstract (**page 1, lines 26-27**) and restructured the sections (**page 10**) to accommodate the different possibility of Cse4 copy numbers. While, fluorescence imaging of CBF3 presents a signal throughout the nuclear region we observed only four copies of Cep3 (part of CBF3). A CBF3 structure also has been resolved by Yan et al. 2018, in which the complex was proposed to exist as a dimer. This translates into four copies of Cep3. Therefore, we find it more suitable to leave all observed Cep3 (CBF3) molecules within a kinetochore model.

(2) The model drawn in Figure 4 makes explicit assumptions about the positioning of the four Cse4 molecules (or two nucleosomes) in each kinetochore relative to the rest of the kinetochore components. Yet, the data shown do not justify this specific arrangement. Lawrimore et al. 2011 claim that the non-centromeric Cse4 nucleosomes must be randomly distributed in the pericentromeric chromatin to evade detection in biochemical tests. Therefore, the nearest-neighbor analysis suggested above will be valuable for gaining new insights into the relative positioning of the centromeric- and non-centromeric Cse4 nucleosomes. A similar analysis for Cep3 and Ctf19 will also be helpful. If stereotypical positioning of these molecules cannot be detected, then the model should be revised accordingly (alternative models that are also consistent with the data can be included).

The reviewer has pointed out that Lawrimore et al. 2011 proposed and justified the existence of a non-centromeric Cse4 pool. This arrangement, also potentially along other inner kinetochore components, makes sense and our data did not indicate it otherwise. Therefore, we now revised our model accordingly by applying changes in the main text on **page 10 lines 302-305** as well as in **Fig. 4**.

(3) I suggest one experiment that can help the authors better understand protein organization in one kinetochore. Joglekar et al. 2006 used a dicentric chromosome to isolate single kinetochores on the spindle axis to test the assumption that each kinetochore consists of approximately the same number of molecules of kinetochore proteins. The strains are easy to construct (transform existing strains with a linearized plasmid). Single kinetochores can be seen with a low but reasonable frequency. I leave the decision to perform the experiment to the authors' discretion depending on whether the experiment will be worth the effort in strengthening or enhancing their conclusions.

We performed the suggested experiment using the strain published in Joglekar et al. 2006 (kindly provided by Prof. Kerry Bloom) with Cse4 additionally tagged with mMaple. However, we always observed several super-resolved Cse4 clusters (likely of several kinetochores) overlapping with Nuf2-GFP diffraction-limited signal, therefore unable to assign a single isolated kinetochore to the lagging centromere.

5. Information regarding the degree of correction applied to calculate protein count per half-spindle: It will be helpful to include data regarding the degree of correction applied to the expected and measured numbers of NPC protein as supplementary data so that the readers can see the magnitude of this correction relative to the measured counts.

We would like to clarify that we did not correct the data. Instead, we calibrate the copy number, given that the copy number of Nup188 per NPC is known. We assume the same ratio between localization and copy number applies to both Nup188 and the kinetochore proteins. We now include a new **Table S4** listing calibration factors of all experiments shown in **Fig. 3**.

Minor points:

1. McIntosh et al. JCB 2013 used microtubule plus-ends in serial section electron micrographs of yeast spindles to align the centromeric region and found a disk-shaped structure that roughly corresponds to the size of a single nucleosome ~ 80 nm away from the tip of the microtubule and centered the microtubule axis. The authors should refer to this finding in their discussion of the model that they present with two nucleosomes. In my opinion, this is compelling evidence for a nucleosome-like structure serving as the kinetochore foundation.

We agree with this reviewer's comment. The study, among others, present compelling evidence for a point-centromere. We now included the finding in the discussion on **page 10, lines 293-294**.

2. As discussed by the authors, the number of Cse4 molecules per kinetochore has been the subject of some controversy. Biochemical data from the Biggins group and ChIPseq data from the Westermann group (Altunkaya et al. 2016 Current Biology) strongly suggest that Cse4 molecules can only be found centered on the centromeric sequence. The latter reference should be included in the discussion.

Thank you for pointing this out. Indeed, this is important. We have now added the relevant reference in the discussion on **page 10 lines 291-292**.

3. Although microscopy-based methods have estimated anywhere from 1, 2, to 6 Cse4 molecules per kinetochore, these studies generally agree on the stoichiometry between Cse4 and the rest of the kinetochore proteins, e.g., Ndc80 complex proteins are ~ 4-fold more abundant than Cse4, etc. The present study seems to disagree with protein stoichiometry. The authors may find it worthwhile to note this feature of their data.

We now discuss the stoichiometry difference between our results and others on **page 11 lines 322-324**.

4. Omission of the Dam1 complex from this study is disappointing to me personally, but I am sure that the authors have good reasons for this. They should briefly comment on the absence of the Dam1 complex in this study.

To provide information on the Dam1 complex, we imaged Ask1, a component of the complex. The measured positioning and copy number of the protein are now included in **Fig. 2** and **Fig. 3** respectively, and described and discussed in respective parts of the manuscript.

Reviewer #1 (Significance (Required)):

Cieslinski and colleagues present a single-molecule localization-based study to define the copy numbers and relative organization of kinetochore proteins in budding yeast. These numbers confirm and significantly refine prior measurements of the same aspects of the kinetochore. They also raise new questions and point to new research directions. The measurements also reveal a model of the protein organization of the budding yeast kinetochore in metaphase. For these reasons, the manuscript is of significant interest to the cell division field.

Reviewer #2 (Evidence, reproducibility and clarity (Required)):

In this study, Cielinski and colleagues have applied single molecule localization microscopy to map the positions of proteins in the yeast kinetochore. This has not been reported previously and this study is both well-conducted and the data appear solid. They also use a modification of this technique to assess the stoichiometry of kinetochore proteins. The results that they obtain are broadly in line with several previous studies that use other methodology. There may be an improvement in accuracy using this new approach that has not been obtained previously and there are some important novel conclusions from this work. I would like the authors to address the following concerns prior to publication:

Major points

1. One interesting finding is that there is a discrepancy in the length of both the MIND and NDC80 complexes (from crystallographic data) with their relative positions. The authors suggest that the outer complexes could be twisted or rotated in respect of the spindle axis. It would be great if the authors could illustrate this in their model (or discuss it in the text), to demonstrate the required angle of twist/rotation of both complexes to account for the discrepancy. A twisted filament structure to the outer kinetochore does have some implications for its response to tension - a key determinant of kinetochore-microtubule attachment. It also may provide some flexibility to the structure under tension.

The discussion about this discrepancy has now been incorporated in the main text, page 9 lines 263-267. For clarity, we only partially reflect this in our schematic model (Fig. 4A; the MIND complex) but we already reflected this in the illustrative structural model in Fig. 4B.

2. For the experiment with cycloheximide, the authors state "Although we observed minor changes in copy numbers, the overall effect of CHX was small." For some proteins, Cse4i for example, there appears to be a significant decrease in intensity (30-40%) after cycloheximide treatment, see Figure S3. While the conclusion that tag maturation does not affect copy number measurements is sound, I suggest modifying this section to reflect the data.

We now modified the section accordingly by pointing out that Cse4i under CHX measurements led to

reduction of the signal. The modification can be found on **page 8 lines 207-211**.

3. Page 5. The statement "These data agree reasonably well with previous diffraction-limited dual-color microscopy studies ..." provides readers with little ability to compare the data. I would like to see a supplementary figure comparing these new data with previous studies, especially those of Joglekar et al 2009, see Figure 3 in this paper.

We thank the reviewer for suggesting such a table. This will allow readers a direct comparison of the data between our study and Joglekar et al. 2009. The comparison can be found in new **Table S1** and **Fig. S4**, which are now mentioned on **page 5**.

4. In terms of the distances quoted, are they in one dimension (as per Jogelkar et al 2009) or in three? The results section is entitled "...positions of kinetochore proteins along the metaphase spindle axis", which suggests a single dimension. Please make this very clear in the results section. In the discussion, is the statement "we mapped the relative positions of 15 kinetochore proteins along the kinetochore axis", which is not entirely clear. It seems from the methods that this is one dimension "...we determined the average distance between the two proteins along the spindle axis. "I suggest clarifying the results section briefly and clearly to indicate that this is a single dimension being measured and also using consistent wording of the axis measured throughout the text.

We agree the previous description may not be clear to the viewers. We now changed the text accordingly in the results section, **page 5 lines 129-130**.

Minor points:

Abstract: I would drop "all" from "For all major kinetochore proteins...", since full characterisation was performed on 14 proteins (9 in terms of copy number).

We now deleted "all" in the abstract as the reviewer suggested.

Page 2: "trough" to through.

Corrected.

Page 2 "S. cerevisiae" to italics

Corrected.

Methods p11. How do the MKY strains relate to common yeast genetic backgrounds? (e.g. are they S288C?).

MKY strains are derivative of S288C. The information was now updated in the Methods section and in **Table S2**.

Reviewer #2 (Significance (Required)):

This manuscript, together with an accompanying one from Virat et al., are nice complementary studies that provide the first single molecule localization studies of the yeast kinetochore. Although other labs have used super-resolution methods to study individual kinetochore proteins; both of these new studies map distances between many proteins at the kinetochore and thus are able to produce maps of the overall kinetochore structure. Like the previous study using standard resolution methods (Joglekar et al, 2009. *Current Biology* 19, 694-699); these studies will likely provide a benchmark for future studies on eukaryotic kinetochore architecture, including those in mammalian systems. Additionally, this work will appeal to super-resolution microscopists.

My expertise is as a yeast kinetochore cell biologist.

November 21, 2022

RE: JCB Manuscript #202209094T

Dr. Jonas Ries
European Molecular Biology Laboratory
Meyerhofstr. 1
Heidelberg 69117
Germany

Dear Dr. Ries,

Thank you for submitting your revised manuscript entitled "Nanoscale structural organization and stoichiometry of the budding yeast kinetochore." We would be happy to publish your paper in JCB pending the minor textual changes recommended by the reviewers as well as final revisions necessary to meet our formatting guidelines (see details below).

A. MANUSCRIPT ORGANIZATION AND FORMATTING:

1) Text limits: Character count for Reports is < 20,000, not including spaces and there should be a single combined 'Results and Discussion' section. Count includes title page, abstract, introduction, results, discussion, and acknowledgments. Count does not include materials and methods, figure legends, references, tables, or supplemental legends.

2) Figure formatting: Reports may have up to 5 main text figures. Scale bars must be present on all microscopy images, including inset magnifications.

3) Statistical analysis: Error bars on graphic representations of numerical data must be clearly described in the figure legend. The number of independent data points (n) represented in a graph must be indicated in the legend. Statistical methods should be explained in full in the materials and methods. For figures presenting pooled data the statistical measure should be defined in the figure legends. Please also be sure to indicate the statistical tests used in each of your experiments (both in the figure legend itself and in a separate methods section) as well as the parameters of the test (for example, if you ran a t-test, please indicate if it was one- or two-sided, etc.). Also, if you used parametric tests, please indicate if the data distribution was tested for normality (and if so, how). If not, you must state something to the effect that "Data distribution was assumed to be normal but this was not formally tested."

4) Materials and methods: Should be comprehensive and not simply reference a previous publication for details on how an experiment was performed. Please provide full descriptions (at least in brief) in the text for readers who may not have access to referenced manuscripts. The text should not refer to methods "...as previously described."

5) For all cell lines, vectors, constructs/cDNAs, etc. - all genetic material: please include database / vendor ID (e.g., Addgene, ATCC, etc.) or if unavailable, please briefly describe their basic genetic features, even if described in other published work or gifted to you by other investigators (and provide references where appropriate). Please be sure to provide the sequences for all of your oligos: primers, si/shRNA, RNAi, gRNAs, etc. in the materials and methods. You must also indicate in the methods the source, species, and catalog numbers/vendor identifiers (where appropriate) for all of your antibodies, including secondary. If antibodies are not commercial, please add a reference citation if possible.

6) Microscope image acquisition: The following information must be provided about the acquisition and processing of images:

- a. Make and model of microscope
- b. Type, magnification, and numerical aperture of the objective lenses
- c. Temperature
- d. Imaging medium
- e. Fluorochromes
- f. Camera make and model
- g. Acquisition software
- h. Any software used for image processing subsequent to data acquisition. Please include details and types of operations involved (e.g., type of deconvolution, 3D reconstitutions, surface or volume rendering, gamma adjustments, etc.).

7) References: There is no limit to the number of references cited in a manuscript. References should be cited parenthetically in the text by author and year of publication. Abbreviate the names of journals according to PubMed.

8) Supplemental materials: There are strict limits on the allowable amount of supplemental data. Reports may have up to 3 supplemental figures and 10 videos. You currently exceed this limit and while we may be able to give you extra space if necessary please try to consolidate these. Figures can take up a full length page. Since you only have 4 main figures you may also move some of the supplemental data to a new main figure.

9) eTOC summary: A ~40-50 word summary that describes the context and significance of the findings for a general readership should be included on the title page. The statement should be written in the present tense and refer to the work in the third person. It should begin with "First author name(s) et al..." to match our preferred style.

10) Conflict of interest statement: JCB requires inclusion of a statement in the acknowledgements regarding competing financial interests. If no competing financial interests exist, please include the following statement: "The authors declare no competing financial interests." If competing interests are declared, please follow your statement of these competing interests with the following statement: "The authors declare no further competing financial interests."

11) A separate author contribution section is required following the Acknowledgments in all research manuscripts. All authors should be mentioned and designated by their first and middle initials and full surnames. We encourage use of the CRediT nomenclature (<https://casrai.org/credit/>).

12) ORCID IDs: ORCID IDs are unique identifiers allowing researchers to create a record of their various scholarly contributions in a single place. At resubmission of your final files, please consider providing an ORCID ID for as many contributing authors as possible.

B. FINAL FILES:

Thank you for this interesting contribution, we look forward to publishing your paper in Journal of Cell Biology.

Sincerely,

Arshad Desai, PhD
Monitoring Editor
Journal of Cell Biology

Dan Simon, PhD
Scientific Editor
Journal of Cell Biology

Reviewer #1 (Comments to the Authors (Required)):

Lando and colleagues use single molecule localization microscopy to reveal the architecture and stoichiometry of kinetochore proteins in budding yeast. Their data present definitive measurements of both aspects, resolving some of the incongruencies in the results of earlier studies of the budding yeast kinetochore and yielding new insights. Along with the companion study by Virant et al., this manuscript establishes a milestone in our progress in understanding the architecture of the eukaryotic kinetochore.

The revised manuscript has thoroughly addressed all the issues that I raised earlier. I have only two comments.

1. A typo on line 283 - "Cef3"
2. In retrospect, the authors' observation that following cycloheximide treatment, the copy number for Cse4 reduces from 4 to ~ three is quite interesting. This decrease could be supportive of prior observations that Cse4 from non-centromeric loci is continuously turned over and degraded. The inhibition of protein synthesis may tip the balance toward enhanced removal of pericentromeric Cse4. The authors should consider citing and discussing the following two studies: Collins et al. Current Biology 2004 (Biggins lab) and Krassovsky et al. PNAS 2011 (Henikoff lab).

Reviewer #2 (Comments to the Authors (Required)):

The authors have addressed my original concerns and I now feel that this publication is suitable for publication. Specifically, the authors have addressed the discrepancy in the lengths of both the MIND and NDC80 complexes (in Figure 4 and the text). They have pointed out that cycloheximide does decrease Cse4 signal. They have included a supplementary table and figure to compare their data with previous data. Finally, the authors have clarified the dimensions in which the measurements are made.

Reviewer #1 (Comments to the Authors (Required)):

~~Cieslinski Lande~~ and colleagues use single molecule localization microscopy to reveal the architecture and stoichiometry of kinetochore proteins in budding yeast. Their data present definitive measurements of both aspects, resolving some of the incongruencies in the results of earlier studies of the budding yeast kinetochore and yielding new insights. Along with the companion study by Virant et al., this manuscript establishes a milestone in our progress in understanding the architecture of the eukaryotic kinetochore.

We thank Reviewer 1 for the appreciation.

The revised manuscript has thoroughly addressed all the issues that I raised earlier. I have only two comments.

1. A typo on line 283 - "Cef3"

The typo is now corrected.

2. In retrospect, the authors' observation that following cycloheximide treatment, the copy number for Cse4 reduces from 4 to ~ three is quite interesting. This decrease could be supportive of prior observations that Cse4 from non-centromeric loci is continuously turned over and degraded. The inhibition of protein synthesis may tip the balance toward enhanced removal of pericentromeric Cse4. The authors should consider citing and discussing the following two studies: Collins et al. Current Biology 2004 (Biggins lab) and Krassovskiy et al. PNAS 2011 (Henikoff lab).

*We now cited and discussed the two suggested studies on **Page 9 lines 235-237**.*

Reviewer #2 (Comments to the Authors (Required)):

The authors have addressed my original concerns and I now feel that this publication is suitable for publication.

Specifically, the authors have addressed the discrepancy in the lengths of both the MIND and NDC80 complexes (in Figure 4 and the text). They have pointed out that cycloheximide does decrease Cse4 signal. They have included a supplementary table and figure to compare their data with previous data. Finally, the authors have clarified the dimensions in which the measurements are made.

We thank Reviewer #2 for the appreciation.